# CHIKV infection reprograms codon optimality to favor viral RNA translation by altering the tRNA epitranscriptome

Jennifer Jungfleisch[1,4], René Böttcher [1✉], Marc Talló-Parra[1,4], Gemma Pérez-Vilaró[1], Andres Merits[2], Eva Maria Novoa [3] & Juana Díez [1✉]

Ample evidence indicates that codon usage bias regulates gene expression. How viruses, such as the emerging mosquito-borne Chikungunya virus (CHIKV), express their genomes at high levels despite an enrichment in rare codons remains a puzzling question. Using ribosome footprinting, we analyze translational changes that occur upon CHIKV infection. We show that CHIKV infection induces codon-specific reprogramming of the host translation machinery to favor the translation of viral RNA genomes over host mRNAs with an otherwise optimal codon usage. This reprogramming was mostly apparent at the endoplasmic reticulum, where CHIKV RNAs show high ribosome occupancy. Mechanistically, it involves CHIKV-induced overexpression of KIAA1456, an enzyme that modifies the wobble U34 position in the anticodon of tRNAs, which is required for proper decoding of codons that are highly enriched in CHIKV RNAs. Our findings demonstrate an unprecedented interplay of viruses with the host tRNA epitranscriptome to adapt the host translation machinery to viral production.

[1] Molecular Virology group, Department of Medicine and Life Sciences, Universitat Pompeu Fabra, 08003 Barcelona, Spain. [2] Institute of Technology, University of Tartu, 50411 Tartu, Estonia. [3] Centre for Genomic Regulation (CRG), The Barcelona Institute of Science and Technology, 08003 Barcelona, Spain. [5]These authors contributed equally: Jennifer Jungfleisch, Marc Talló-Parra. ✉email: rene.boettcher@outlook.com; juana.diez@upf.edu

Viruses completely depend on the host translation machinery to express their proteins. This is particularly apparent for positive-strand RNA ((+)RNA) viruses that, upon release of their genomes into the cytoplasm, have first to translate their replicases to establish an infection because they do not carry viral replicases within their virions. Paradoxically, however, genomes of diverse (+)RNA viruses are enriched in rare codons and consequently should be translated with poor efficiency. Redundancy of the genetic code permits amino acids to be encoded by more than one codon. The frequency of these synonymous codons is universally biased and organism-specific. As tRNA concentrations in a given organism are adapted to the codon usage, rare codons slow down the elongation rate and thus decrease translation efficiency because their cognate tRNAs are lowly abundant in comparison to those tRNAs used for optimal codons[1–3].

Given the dependence of (+)RNA viruses on the host translation machinery, one would predict a strong selection pressure for viral genes to adapt to the host codon usage, at least for viruses that highly express their genomes. Indeed, this seems to be the case for viruses that infect bacteria[4,5], however, it is not a general trait for viruses that infect mammals[6–8]. The genes of highly expressed human host mRNAs are enriched in G/C-ending codons whereas the genomes of diverse (+)RNA viruses such as coronaviruses or the emerging Dengue virus (DENV) and Chikungunya virus (CHIKV) are enriched in A/U ending codons[9,10]. How these viruses efficiently translate their genomes in spite of their unfavorable codon usage remains a fundamental question in virology. To address this question, we use CHIKV as a model virus because its (+)RNA genome is translated and replicated at extremely high levels[11].

In this work we show how CHIKV infection induces a codon-specific reprogramming of the host translation machinery to favor viral protein expression via alterations of the host tRNA epitranscriptome.

## Results

**CHIKV infection specifically alters the translational landscape at the endoplasmic reticulum.** The CHIKV (+)RNA genome consists of a 5′capped and 3′poly(A)-tailed single-stranded RNA that contains two open reading frames (ORFs). The first ORF encodes four non-structural proteins required for RNA replication. The second ORF is expressed from a subgenomic (SG) RNA transcribed during infection and encodes five structural proteins present in the virion (Fig. 1a)[12]. To investigate how CHIKV efficiently translates its genomic and SG RNAs, we performed global translatome profiling of both endoplasmic reticulum (ER) and cytosol fractions from non-infected and CHIKV-infected human HEK 293T cells. Cells were infected at a multiplicity of infection (MOI) of 4 and samples collected at twelve hours post-infection, a time of high viral production and no cytopathic effect (Supplementary Fig. 1a–c). We separately analyzed effects at the ER and the cytosol, two distinct biological environments for translation whose contribution to viral RNA translation is mainly unexplored. Proper subcellular fractionation was confirmed (Supplementary Fig. 1d) and corresponding libraries for RNA-seq and ribosome profiling (Ribo-seq) were generated. After initial quality assessment of both sequencing techniques and removal of low-quality samples (see Methods), our RNA-seq analysis showed that CHIKV infection caused a massive decrease of cellular mRNA levels in both translation compartments (Fig. 1b) while solely a small group of genes enriched for immune response functions were up-regulated (Supplementary Data 1). This drastic decrease of cellular mRNA levels was not surprising as CHIKV nsP2 mediates the degradation of Rpb1, a subunit of the RNA Polymerase II (RNAPII)[13]. A virus-

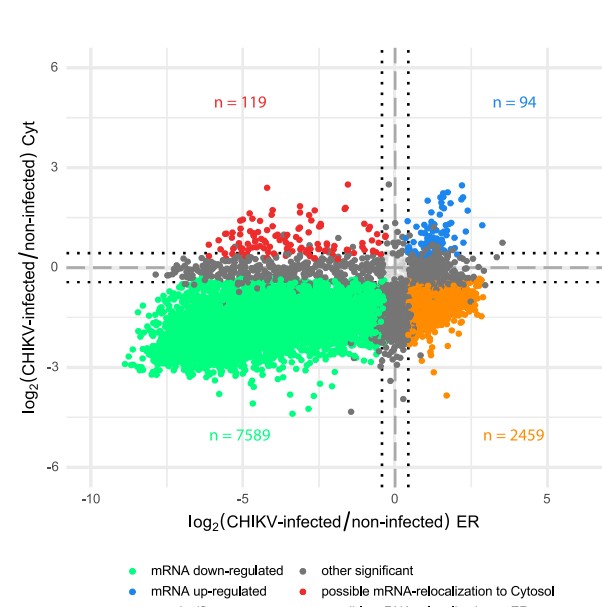

**Fig. 1 CHIKV infection alters the cell host transcriptome. a** Scheme of CHIKV RNA genome. **b** Transcriptome analyses of the cytosol (Cyt) and ER compartments in CHIKV-infected cells. Genes were plotted according to their $\log_2$ fold changes in CHIKV infected *vs.* non-infected cells for both translation compartments. The sequencing reads were analyzed using a custom transcriptome annotation and differential expression was determined via limma-voom (see Methods). Colors indicate genes that are significantly (FDR < 0.05) down-regulated in both compartments (green), up-regulated in both compartments (blue), up-or down-regulated in only one compartment (dark grey) or show opposing behavior in the two compartments (orange and red). Dotted lines indicate $\log_2$ fold changes of ±0.433 (i.e. a 35% change in either direction) used as minimum threshold to declare mRNA levels as changed. HEK 293T cells were infected at a MOI of 4 and samples collected at twelve hours post-infection. Three independent replicates were carried for each condition.

induced host RNA decay might also contribute to the effect as Sindbis, a closely related virus, upregulates host mRNA decay[14]. Of note, some mRNAs exhibited an opposing behavior in the two compartments suggesting that CHIKV infection might induce specific mRNA shuttling. The mRNAs up-regulated in the cytosol and down-regulated at the ER were functionally enriched in gene ontology (GO) terms related to membranes, exocytosis, lysosomes, protein maturation and ER stress. By contrast, the mRNAs that were up-regulated at the ER and down-regulated in the cytosol were enriched in GO terms related to metabolism, mitochondrial organization and translational elongation (Supplementary Data 2). Expression levels of a selected set of mRNAs was validated by qPCR in an independent experiment (Supplementary Fig. 2).

A comparison of viral and host reads in the RNA-seq and Ribo-seq libraries indicated that CHIKV RNAs exhibited a far higher ribosome occupancy at the ER than at the cytosol (Fig. 2a, Supplementary Fig. 3). The percentages of viral ribosome protected fragments (RPFs) and RNA-seq reads in the cytosol were similar whereas the percentage of viral RPFs at the ER was much higher than expected when compared to that of viral RNA-seq reads. Here, one might argue that an important fraction of the RNA-seq reads of CHIKV RNAs might correspond to viral RNAs

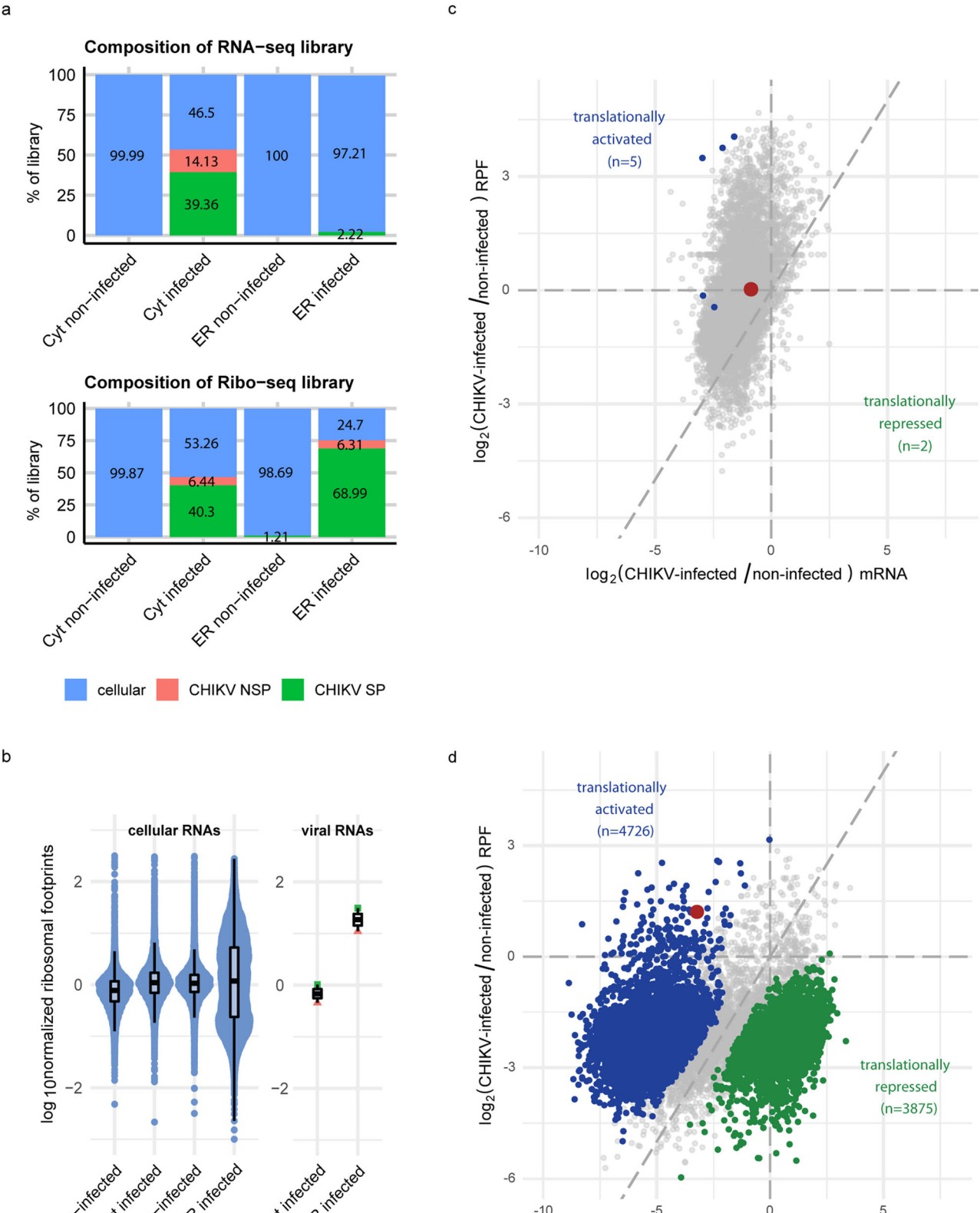

destined for replication and encapsidation, thereby confounding our analyses. However, in such a case we would expect to see substantial differences between the two CHIKV ORFs, since the NSP ORF is translated from the genomic CHIKV RNA, that acts as an mRNA and as a template for replication and encapsidation, whereas the SP ORF is translated from the subgenomic CHIKV

RNA that mainly acts as an mRNA. As both CHIKV ORFs showed very similar behavior in our analysis, any potential biases seem negligible for our conclusions although we cannot eliminate an underestimation of the ribosome occupancy values for the NSP ORF. Regarding the cellular host mRNAs, when inspecting ribosome occupancy before and after normalization

**Fig. 2 CHIKV reshapes the translational landscape at the ER but not the cytosol. a** Composition of RNA-seq and Ribo-seq libraries from the cytosol and ER compartments in non-infected and CHIKV-infected cells. Displayed are the percentages of total cellular and viral counts (for NSP - non-structural, and SP - structural proteins) after converting raw counts to counts per million (CPM) and averaging the CPM per condition across all samples passing QC. Colors indicate cellular mRNAs (blue), viral genomic RNA encoding non-structural proteins (red) and viral SG RNA encoding structural proteins (green). The percentage of viral NSP RNA-seq counts from the ER compartment in CHIKV-infected cells was too low (0.57%) to be visualized in the figure. **b** Normalized ribosome footprints of cellular mRNAs ($n = 12220$ mRNAs) and viral RNAs ($n = 2$) in both translation compartments in non-infected and CHIKV-infected cells. Normalized ribosome footprints were calculated for each gene (dots) by dividing the mean RPF CPM by mean mRNA CPM and subsequent $\log_{10}$ transformation. Colors as in (**a**). Upper and lower limits of the box correspond to the 1st and 3rd quartile score distribution with whiskers extending to 1.5 times the range from top/bottom of the box. The line represents the median. **c** and **d** Analyses of RNA-seq and RPF libraries comparing CHIKV-infected and non-infected cells in the (**c**) cytosolic and (**d**) ER fractions. Each dot represents one gene, with a red dot highlighting KIAA1456 (which will be discussed in the next sections). Colors indicate no significant changes (grey), significantly increased translation rate (blue), significantly decreased translation rate (green). HEK293T cells were infected at a MOI of 4 and samples collected at twelve hours post-infection. Three independent replicates were carried for each condition.

to the total observed RPFs, the occupancy profiles did not exhibit any marked differences across the CDS (Supplementary Fig. 3a, b), while no evidence for ribosome stalling could be found at either translation initiation or translation termination site (Supplementary Fig. 3c–f). These observations indicate that ribosomes are not stalled and that translation of host mRNAs still occurs, albeit at lower levels than in non-infected cells. Such a decrease in total RPF levels would correlate with the decrease of mRNA levels, reflecting the previously described shutoff occurring in alphaviruses[15–18].

However, we did note an increased occupancy in the 3'UTR of ER samples after infection (Supplementary Fig. 3b), which may indicate non-RPF contaminants in the library that could lead to inflated RPF counts. We therefore further assessed the quality of our libraries in terms of RNA integrity by calculating the transcript integrity number (TIN)[19] for all transcripts across all RNA-seq samples (Supplementary Fig. 4a) and noted markedly lower TIN scores for infected ER samples. Since RNA decay rates can vary between functional groups[20,21], it might be that ER-associated transcripts exhibit higher decay rates than cytosolic RNAs upon CHIKV infection. However, this would not explain an increased ribosome occupancy specific to the 3'UTR. Thus, to investigate the possibility of non-RPF contaminants that may be introduced by co-sedimentation or binding by viral RNA-binding proteins[22], we calculated the normalized ribosomal footprints for long non-coding RNAs (lncRNAs) before and after infection (Supplementary Fig. 4b). In the case of random binding or co-sedimentation of contaminants, we would expect to see a significant increase of nRFs for lncRNAs in infected ER samples, which was not the case. Instead, we observed this behavior for lncRNAs in the cytosol, where we in turn did not observe a similar 3'UTR anomaly (Supplementary Fig. 3b). As additional validation, we visualized the RPF length distributions of host and viral RPFs across the CDS and both UTRs for all three reading frames (Supplementary Fig. 5–8). While host RPFs in the CDS showed a preference for frame 0 in both cytosol and ER, viral RPFs did not, indicating that these RPFs may indeed be contaminated by non-RPF fragments which could lead to an overestimation of viral RPFs in our Ribo-seq libraries. We therefore focused on the host RPFs and compared their length distributions of CDS and 3'UTR in each compartment. Should non-RPF fragments co-sediment with ribosomes, we would expect to see marked differences in these distributions. Reassuringly this was not the case, indicating that the fragments mapping to 3'UTRs may indeed mostly originate from ribosomes. We did not observe any dominant reading frame in either of the UTRs when comparing across all samples, suggesting that any translation in this region would occur both in- and out-of-frame if at all. Based on this combined evidence, we concluded that the 3'UTR-specific occupancy is likely not caused by contaminants,

but may be the result of an impaired post-termination recycling of 80S ribosomes, as has been described previously[23–25]. In line with this, ABCE1, a key component of the ribosome recycling machinery, was down-regulated after CHIKV infection in both the ER and cytosol translation compartments (ER: RNA log2FC $= -6.56$, RPF log2FC $= -3.21$; cytosol: RNA log2FC $= -1.81$, RPF log2FC $= -1.35$).

We therefore continued our analyses by subsequently calculating the normalized ribosomal footprints (nRF = mean RPF counts / mean RNA-seq counts) of host mRNAs and viral genomic and SG RNAs (Fig. 2b). In the cytosol, CHIKV infection caused a mild nRF increase of host mRNAs, with viral RNAs exhibiting comparable nRF levels. In contrast, at the ER, CHIKV infection caused a bimodal distribution of nRF values of host mRNAs. A subset of mRNAs exhibited very high nRF values, in the range of those of viral RNAs, while another subset of mRNAs displayed low ones. Differential ribosome occupancy was validated in an independent experiment for a selected set of mRNAs (Supplementary Fig. 9).

We then analyzed differential translation in both compartments using an interaction term in limma-voom[26,27] to estimate significant differences between RPF and RNA-seq read counts. This type of analysis assesses whether changes in ribosome occupancy can be solely explained by changes in the corresponding mRNA levels while simultaneously accounting for variance between samples and controlling the false discovery rate (FDR). Genes were plotted according to their $\log_2$ fold changes in mRNA abundance and ribosome occupancy (as measured by RPF) in the cytosol and ER translation compartments. After CHIKV infection, few significant translational changes were observed in the cytosol (Fig. 2c), which may be attributable to a limited statistical power in our dataset. Thus, we focused our attention on the ER, where we observed dramatic changes (Fig. 2d). Enrichment analysis with Gene ontology (GO), Kyoto Encyclopedia of genes and genomes (KEGG) and REACTOME pathway databases showed that translationally activated mRNAs were enriched for genes related to cell cycle, RNA transport and DNA damage response, whereas translationally repressed mRNAs were enriched for genes related to mitochondria, oxidative phosphorylation, ribosomes and RNA translation (Supplementary Data 3). Collectively, these results indicate that CHIKV infection induces profound alterations of the translation landscape at the ER, the compartment at which CHIKV RNAs exhibit very high ribosome occupancy and therefore likely efficiently translates.

**CHIKV infection reprograms codon optimality.** To investigate the mechanism by which CHIKV achieves this high ribosome occupancy at the ER, we examined whether the CHIKV RNAs and the translationally-activated host mRNAs share any common features such as length of codon sequence (CDS) and

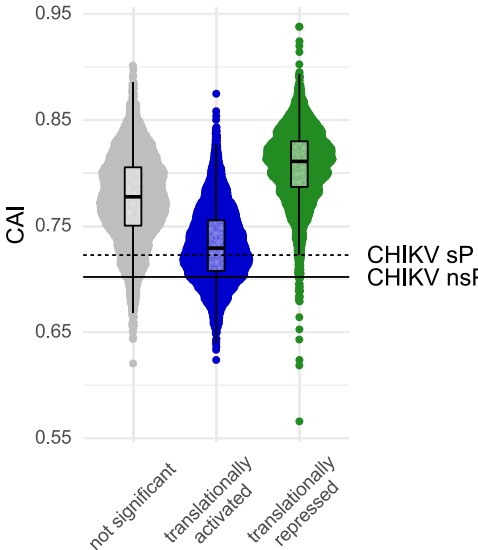

**Fig. 3 CHIKV RNA and translationally activated mRNAs feature low CAI values.** Distribution of codon adaptation index (CAI) among different translational groups at the ER. Each dot represents one gene. Horizontal lines indicate CAI of CHIKV ORFs: continuous line - NSP, dashed line - SP. Upper and lower limits of the box correspond to the 1st and 3rd quartile score distribution with whiskers extending to 1.5 times the range from top/bottom of the box. The line represents the median. Translationally activated mRNAs $n = 4726$, translationally repressed mRNAs $n = 3875$, not significantly changed mRNAs $n = 3619$.

untranslated regions (UTRs), GC content, RNA secondary structure or codon usage bias. Overall, translationally activated host mRNAs tended to be similar to the CHIKV RNAs, featuring a longer CDS with a lower GC content when compared to host mRNAs that were either translationally repressed or not significantly altered (Supplementary Fig. 10). Likewise, these mRNAs were generally also less structured (Supplementary Fig. 11), although a comparison with CHIKV RNAs was not possible since the utilized Parallel Analysis of RNA Structure (PARS) scores were not available for the viral RNA. Codon usage bias was measured via the codon adaptation index (CAI) which measures the synonymous codon bias with respect to a set of reference genes. It is used as a quantitative method to predict the level of protein expression of a gene based on its codon sequence under basal conditions. For instance, a viral ORF sequence with a CAI value of 1 would exclusively contain the most commonly used codons of the host. Remarkably, the CHIKV RNA and the host translationally-activated mRNAs featured low CAI values, whereas the translationally-repressed host mRNAs featured high CAI values (Fig. 3), indicating a virus-induced reprogramming of codon optimality. Codon usage comparisons revealed a clustering of viral RNAs with translationally-activated host mRNAs (Supplementary Fig. 12). To further characterize this behavior, we calculated the mean difference between the codon frequencies of translationally-activated and translationally-repressed (Supplementary Fig. 13), as well as of the CHIKV RNAs and translationally-repressed mRNAs. In both the translationally activated host genes and the CHIKV RNAs we observed that GAA (Glu), AAA (Lys), CAA (Gln) and AGA (Arg) were among the most enriched codons (Fig. 4). Interestingly, their decoding is linked to two modifications at the wobble U34 position of the cognate tRNAs, mcm⁵ (5-methoxy-carbonyl-methyl) and mcm⁵s² (5-methoxy-carbonyl-methyl-2-thio) (reviewed in[28]). Moreover, we also observed an enrichment of GGA (Gly) codons

whose decoding in humans is linked to the modifications mcm⁵ and mchm⁵ (5-methoxy-carbonyl-hydrox-methyl)[29].

CHIKV, as all mosquito-borne viruses, replicates alternatively in vertebrates and mosquitoes, two hosts separated by one billion years in evolution that harbor different codon usage biases. Thus, the suboptimal coding usage of CHIKV RNA in human cells could be driven by a competing selection to adapt to the mosquito codon usage. However, this does not seem to be the case, as the CAI of CHIKV RNA relative to the codon usage of human (CAI = 0.7123) and *Aedes albopictus* (CAI = 0.7113), a known CHIKV vector, are quite similar when considering all annotated protein-coding genes. Moreover, as shown in Fig. 4, the CHIKV codon usage does not show a preference for the same rare codons when compared to *A. albopictus* genes, however, this comparison does not account for any possible virus-induced changes in gene expression or differences in the codon optimality. Since the CAI of CHIKV relative to human translationally activated genes slightly increased to 0.7288 (NSP = 0.7224; SP = 0.7416), global suboptimality of the CHIKV genome might be determined by alternating hosts, whereas preference for some specific codons would be determined by stress-related changes in tRNA modification status.

**KIAA1456 mediates virus-induced reprogramming of codon optimality.** In *Saccharomyces cerevisiae*, Trm9 converts the cm⁵-modified wobble uridine addition into mcm⁵ in a subgroup of tRNAs[30]. Further thiolation of mcm⁵ to mcm⁵s² involves the activity of Urm1, CTU1/2 and NFS1[31–33]. In mammals, *TRM9* has two predicted homologs, *TRM9/ALKBH8* and *TRM9b/TRM9L/KIAA1456*. Both ALKBH8 and KIAA1456 contain an S-adenosyl methionine methyltransferase domain (SAM), but whereas ALKBH8 contains additionally an AlkB domain and an RNA recognition motif (RRM), KIAA1456 contains an internally disordered region harboring several phosphorylation sites[34]. ALKBH8 catalyzes the formation of mcm⁵ in vitro, partially rescues mcm⁵ formation in Trm9-deficient yeast cells and is fully characterized[35,36]. In contrast, KIAA1456 is unable to rescue mcm⁵ formation in Trm9-deficient yeast, suggesting that it might require co-factors absent in yeast. Although KIAA1456 overexpression results in an increase in mcm⁵ modification in a cancer cell line[37], no methyltransferase activity has been demonstrated in vitro yet.

While we did not detect ALKBH8 mRNA in HEK293T cells, TRM9b/TRM9L/KIAA1456 mRNA was amongst the most translationally activated mRNAs in CHIKV-infected cells at the ER (Supplementary Data 1). From here on, we will refer to this mRNA as KIAA1456. During infection, KIAA1456 mRNA levels were downregulated akin to the majority of mRNAs. However, this effect was buffered by a dramatic translational activation (12-fold) that resulted in an absolute increase of RPFs and corresponded to a 1.4-fold increase in KIAA1456 protein expression levels (Supplementary Data 1, Fig. 5a and Supplementary Fig. 14). Liquid chromatography with tandem mass spectrometry (LC–MS/MS) analyses of tRNA isolated from non-infected and CHIKV-infected cells, as well as from non-infected cells overexpressing a myc-DDK-tagged-KIAA1456 protein (Fig. 5b), showed that both CHIKV infection and KIAA1456 overexpression induced a significant increase in the levels of mcm⁵ modification (CHIKV-infection: 1.3-fold change, $p = 0.011$, two-tailed *t*-test; KIAA1456: 1.4-fold change, $p = 0.016$, two-tailed *t*-test) (Fig. 5c). This suggests that KIAA1456, like ALKBH8, catalyzes the required methylation to generate mcm⁵. In agreement with the observed CHIKV-induced translational repression of the four enzymes directing mcm⁵ thiolation (Supplementary Data 4), we did not observe an

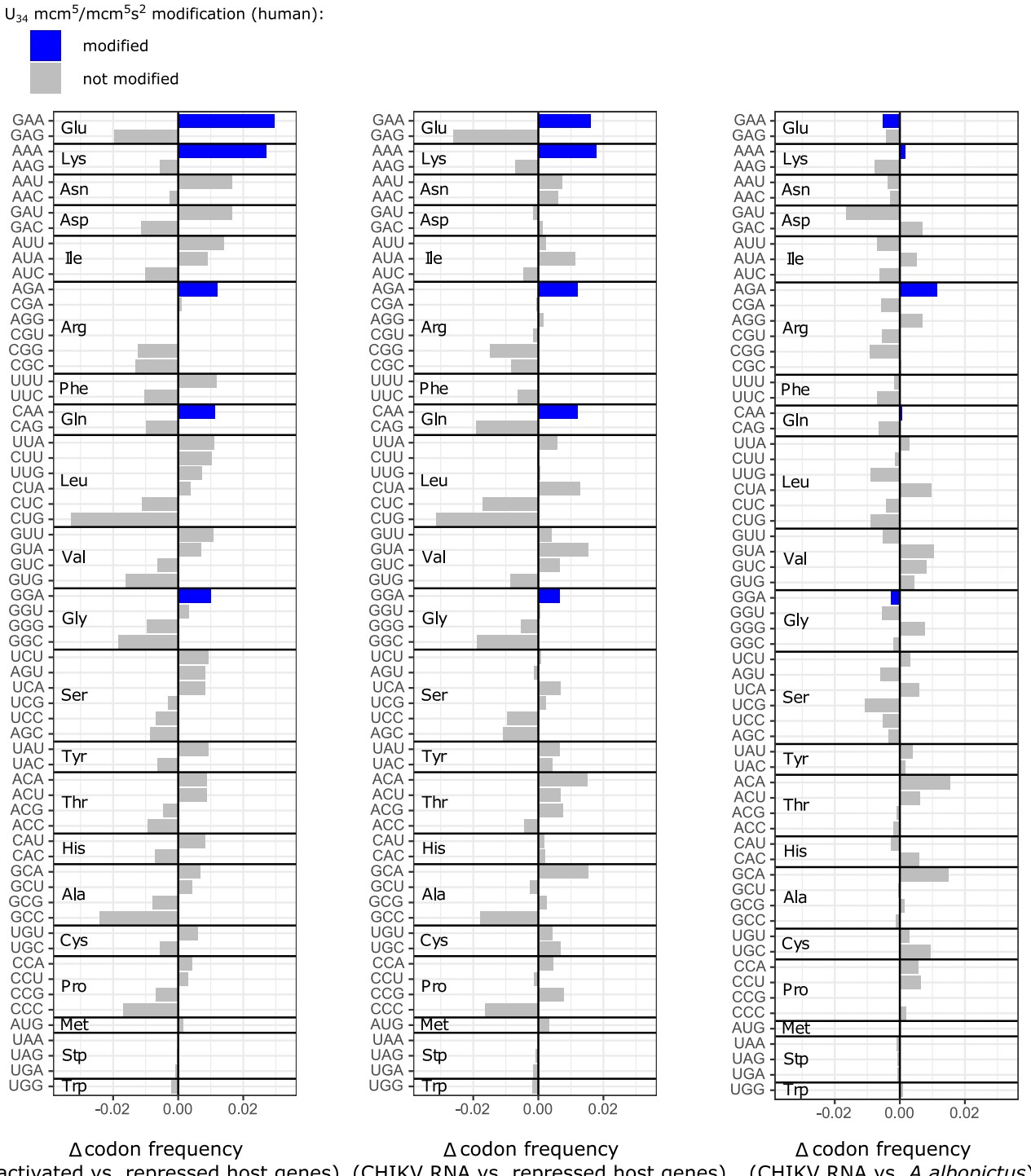

**Fig. 4 CHIKV infection induces codon usage reprogramming in human cells.** Difference in mean codon frequency (see Fig. S13) for translationally activated host genes compared to translationally repressed host genes (left panel), for CHIKV RNAs compared to translationally repressed host genes (middle panel) or for CHIKV RNAs compared to the Asian tiger mosquito *Aedes albopictus*, a known vector (right panel).

increase in mcm5s2 modification levels (Fig. 5c). KIAA1456 overexpression and CHIKV infection caused a similar increase in the levels of tRNA mcm5 modification in spite of that KIAA1456 protein expression levels achieved by overexpression were 12-fold higher than those achieved via CHIKV infection (Fig. 5b). This suggests that steps prior to the KIAA1456 catalyzed methylation might become limiting. One possible limiting step is the acetylation step directly prior to the methylation which is catalyzed by the acetyltransferase complex

Elongator ELP1-6, a complex consisting of six subunits. In line with this, our ribosome profiling data show that the absolute RPF levels of all ELP subunits are downregulated after CHIKV infection (Supplementary Data 1 and 4). As predicted for a positive role of KIAA1456 in CHIKV infection, overexpression of KIAA1456 protein favored CHIKV RNA replication (fold change 1.2, *p*-value = 0.0045, two-tailed *t*-test) (Fig. 5d), and CHIKV protein expression (Fig. 5e). However, we did not detect any effect in virus titers (Supplementary Fig. 15a). This

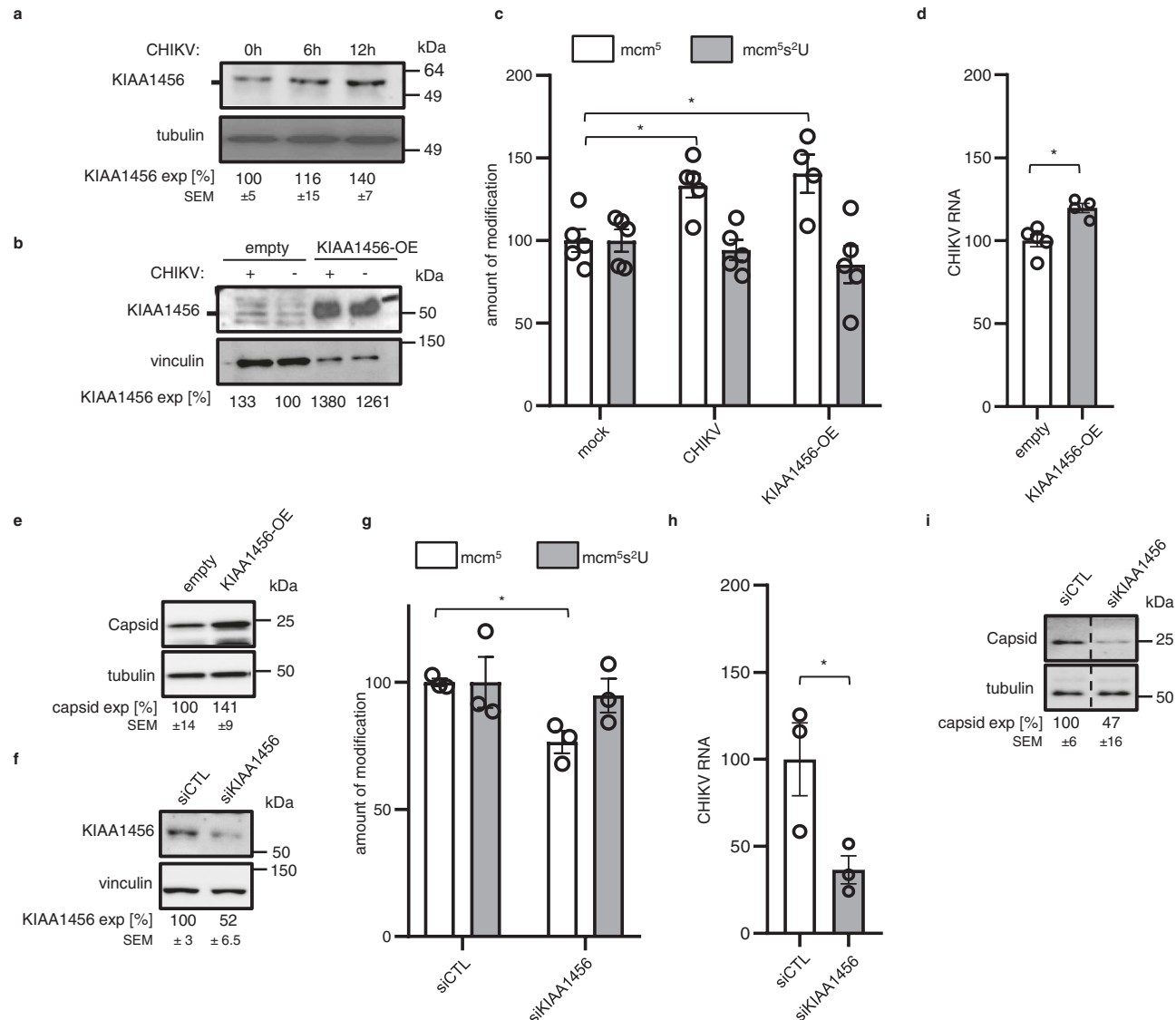

**Fig. 5 Effect of KIAA1456 overexpression and depletion on mcm$^5$U and mcm$^5$s$^2$U modification levels and on CHIKV infection. a** Western blot showing KIAA1456 protein upregulation upon CHIKV infection. Values denote relative KIAA1456 expression normalized by tubulin expression calculated from 3 independent replicates and expressed as mean ± SEM. **b** Western blot showing KIAA1456 protein levels under plasmid-driven overexpression (KIAA1456-OE) and CHIKV infection conditions. Values represent relative KIAA1456 expression normalized by vinculin expression. A representative example from two independent replicates is shown. Note, that to be able to compare expression levels, 1:5 dilutions of the KIAA1456–OE overexpression samples were loaded. **c** Relative mcm$^5$U and mcm$^5$s$^2$U modification levels in mock-infected cells, CHIKV-infected cells and in cells overexpressing KIAA1456 (KIAA1456-OE) determined by mass LC-MS-MS; bars represent the mean value ± SEM of 4 (in the case of mcm$^5$ after KIAA1456-OE) respectively 5 (in the remaining cases) independent replicates. p-values: mcm$^5$ - mock vs CHIKV: 0.011, mcm5 – mock vs KIAA-OE: 0.016. **d** Relative amount of CHIKV RNA levels (p-value 0.0046) and (**e**) of CHIKV capsid protein levels in CHIKV-infected cells transfected with an empty plasmid or with a plasmid overexpressing KIAA1456 (KIAA1456-OE) determined by qPCR; bars represent the mean value ± SEM of 5 (empty) and 4 (KIAA-OE) independent replicates. CHIKV RNA and capsid protein expression levels were normalized by TUBB6 RNA and tubulin, respectively. **f** Western blot showing KIAA1456 protein levels upon transfecting siRNA control (siCTL) and siKIAA1456. Values denote relative KIAA1456 protein expression normalized by vinculin expression calculated from at least 3 independent samples and expressed as mean ± SEM. **g** Relative levels of mcm$^5$U and mcm$^5$s$^2$U modification (p-value 0.007), **h** CHIKV RNA levels (p-value 0.047) and **i** CHIKV capsid protein levels in CHIKV-infected cells, transfected with siCTL or siKIAA1456 determined as in (c, d and e). Bars represent the mean value ± SEM of 3 independent replicates. All CHIKV infections were carried out at an MOI of 4 and samples were collected 12 hpi. All statistical analyses were performed using the two-tailed t-test. *p < 0.05.

might be explained by the lower sensitivity of the plaque assay method in comparison with Western blot and qPCR methods. Alternatively, the amount of CHIKV RNA and protein produced in the infected cell might be in excess in relation to the production of infectious viral particles. Next, we analyzed the effect of the depletion of KIAA1456 in CHIKV infection.

Consistently with a role of KIAA1456 in mcm$^5$ modification and thereby viral RNA translation, siRNA-mediated silencing of KIAA1456 (Fig. 5f) decreased mcm$^5$ levels (Fig. 5g), inhibited CHIKV RNA and protein levels (Fig. 5h, i) and at a low MOI hindered infectious viral particle production (Supplementary Fig. 15b).

**a**

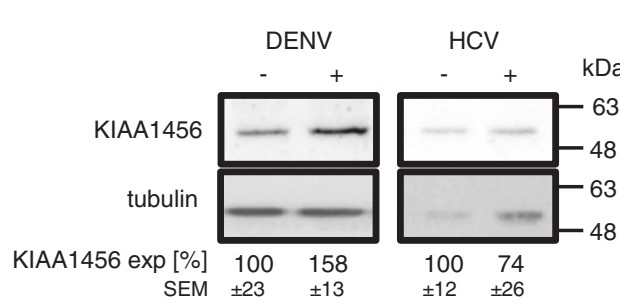

**b**

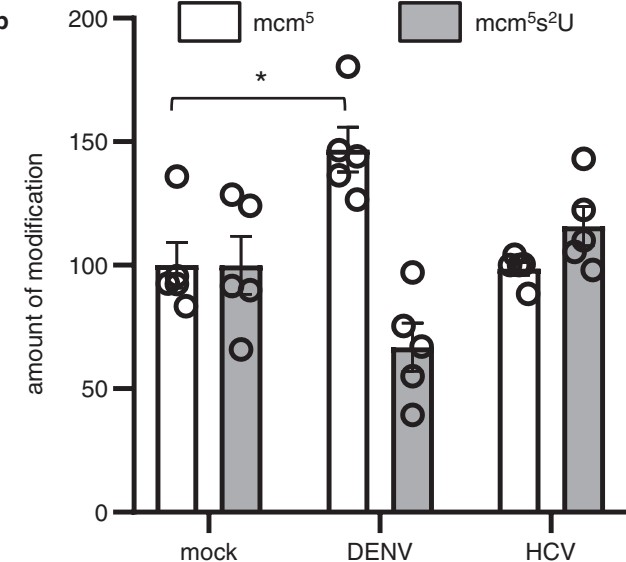

**Fig. 6 Effect of DENV and HCV infection on KIAA1456 protein levels and mcm⁵ and mcm⁵s²U levels. a** Western blot showing KIAA1456 protein expression levels upon DENV and HCV infection. Values denote KIAA1456 expression normalized by tubulin expression relative to uninfected cells calculated from 3 independent replicates and expressed as mean ± SEM. **b** as in Fig. 5c but upon DENV and HCV infection. Bars represent the mean value ± SEM of 5 independent replicates (*p*-value 0.007). Statistical analyses were performed using the two-tailed *t*-test. *$p < 0.05$.

To investigate whether other RNA viruses might also use a KIAA1456-mediated tRNA modification reprogramming as a mechanism to enhance viral RNA translation, we analyzed viral sequences from diverse viruses for GAA, AAA, CAA, AGA and GGA codon enrichment. We observed that multiple (+)RNA viruses, such as DENV, showed an enrichment in these codon subsets, whereas other viruses such as HCV did not (Supplementary Data 5). In agreement with these observations, infection by DENV but not with HCV induced a statistically significant increase ($p = 0.097$ vs. $p = 0.69$) in KIAA1456 protein expression (Fig. 6a). Moreover, this upregulation also coincided with an increase in mcm⁵ modification levels of tRNAs in DENV-infected (fold change 1.8, *p*-value = 0.035, two-tailed *t*-test) but not HCV-infected cells (Fig. 6b). Collectively, these results indicate a mechanistic link between viral RNA enrichment in GAA, AAA, CAA, AGA and GGA codons, virus-induced increase in KIAA1456 levels and elevation of the tRNA mcm⁵ modification at the wobble U34 position.

## Discussion

Our study identifies an unanticipated adaptation of the viral codon usage to the host translation environment under infection conditions. This adaptation occurs through a reprogramming of the host tRNA epitranscriptome that involves the tRNA modifying enzyme KIAA1456. Interestingly, the KIAA1456 mRNA, one of the most translationally activated mRNAs in CHIKV-infected cells, is itself enriched in the GAA, AAA, CAA, AGA and GGA codons. This suggests a positive feedback loop that might be triggered by the massive CHIKV-induced decrease of mRNA levels and the subsequent imbalance between mRNA and tRNA levels. We hypothesize that this imbalance especially affects the ER translation compartment favoring translation of KIAA1456 mRNA and the observed codon usage changes. Two main observations support this idea. First, our RNA-seq data show higher decreases of mRNA levels at the ER translation compartment than at the cytosol one (Fig. 2 c, d). Second, our Ribo-seq data show that KIAA1456 mRNA preferably translates at the ER (Fig. 2 c, d and Supplementary Data 1). This would result in a higher concentration of KIAA1456 protein around the ER that ultimately would favor translation of viral and host mRNAs enriched in GAA, AAA, CAA, AGA and GGA.

CHIKV-infection caused an increase in the levels of tRNA mcm⁵ but not of mcm⁵s² modifications. Indeed, a similar observation was obtained by overexpressing KIAA1456 under translational stress conditions in the highly tumorigenic cells SW620[37]. Our interpretation is that overexpression of KIAA1456 results in mcm⁵ modification of previously unmodified tRNAs. If the Urm1, CTU1/2 and NFS1 enzymes required for thiolation would be limiting, the percentage of mcm⁵s²-modified tRNAs would be conserved. Moreover, the fact, that these enzymes are downregulated after CHIKV infection and that we do not observe an increase of mcm⁵s² suggests that in humans the mcm⁵ modification is sufficient to favor translation of these A-ending codons and that mcm⁵ and mcm⁵s² modifications act cooperatively. This has been previously proposed[38] and goes in line with the fact that *Saccharomyces cerevisiae* cells lacking either mcm⁵ or s² are viable, whereas their simultaneous loss is lethal[39]. However, further studies dissecting the roles of the different U34 modifications on decoding in human cells will be required to understand their impact on the human translatome and to complete the picture of the tRNA epitranscriptome function in infections and disease.

Coordinated interplays between the tRNA epitranscriptome and biased codon usage have been previously described in response to stress conditions to favor expression of codon-biased transcripts expressing stress proteins[40–43]. For example, in yeast, alkylation stress increases the dependence on Trm9 for the expression of GAA- and AGA- enriched DNA damage and cell cycle control genes which are essential for surviving alkylation stress[40]. These two codons are among the most enriched in CHIKV RNAs as well as in CHIKV-induced translationally activated host mRNAs whose GO terms include "cell cycle" and "DNA damage response" (Supplementary Data 3). Together, this is in agreement with *KIAA1456* being a *TRM9* functional homolog and would suggest that viruses might have hijacked components of these highly conserved cell-defense mechanisms to their benefit. A stress response triggered by viral infections would result in a host reprogramming of codon optimality that ultimately would favor translation of viral RNA genomes adapted to this stress environment. Indeed, the use of *TRM9*-related enzymes by viruses seems to be evolutionary conserved beyond human viruses. In *Escherichia coli* the tRNA^Lys U34 modification is essential for the lambda phage to express two crucial proteins

for viral replication, gpG and gpGT. However, in this case the tRNA modification influences translation not by globally reprogramming the codon usage but by affecting a specific ribosomal frameshift required to express the proper gpG:gpGT ratio[44,45].

The knowledge on human tRNA modification genes is fragmentary and only 48% of candidate tRNA modification enzymes have been experimentally validated in mammals[28]. Besides the described enrichment of GAA, AAA, CAA, AGA and GGA codons in CHIKV RNA and in infection-induced translationally activated mRNAs, other A/T-ending codons were also observed to be significantly overrepresented (Fig. 4). These include codons whose recognition by the cognate tRNAs would require a modification at the U34 position yet to be determined[46]. Thus, it seems likely that other, still unknown tRNA modifying enzymes contribute to the observed virus-induced translational reprogramming. Further advances in our understanding of the human tRNA modifying enzymes and of the effect of the corresponding modifications in decoding will be essential to identify them.

In summary, we propose that codon-specific reprogramming of translation via tRNA modification represents a general mechanism used by some viruses to enhance viral protein expression. Besides shedding light on a novel layer of virus–host cell interactions, our results provide a rationale to consider the regulation of the tRNA epitranscriptome as a promising therapeutic target. As neurodegenerative diseases and cancers show alterations in tRNA modification enzymes[47], including *KIAA1456*[37,48,49], the therapeutic interests reach well beyond virology.

## Methods

**Cell lines**. HEK 293T (ATCC; CRL-11268; female), Huh7 (kindly provided by Francis Chisari) and BHK-21 (kindly provided by Andres Merits) cells were grown at 37 °C and 5% CO$_2$. HEK 293T and Huh7 cells were cultivated in Dulbecco's modified Eagle's medium (DMEM, ThermoFisher, 41966-029) supplemented with 10 % fetal bovine serum (FBS) and 1% non-essential amino acids. BHK-21 cells were cultivated in Glasgow Minimum essential medium (GMEM, Lonza, BE12-739F) supplemented with 10% FBS, 10% triptose phosphate buffer and 1 M Hepes pH 7.2.

**Viruses and infection conditions**. Stocks of CHIKV LR2006-OPY1 (GenBank: DQ443544, kindly provided by Andres Merits) and DENV (DENV-2 strain 16681, Genbank Accession NC 001474, kindly provided by Ralph Bartenschlager) were generated in BHK-21 cells and titered by standard plaque assay in HEK 293T. Stock of HCV (HCV JFH1/D183, kindly provided by Francis Chisari) was generated in Huh7 cells and tittered by standard TCID50. CHIKV and DENV infections were carried out in HEK 293T cells with an MOI of 4 or 0.000004 (when indicated) and 1 h of incubation time in the case of CHIKV and with an MOI of 3 and 4 h of incubation time in the case of DENV. HCV infection was carried out in Huh7 cells with an MOI of 3 and 4 h of incubation time. Cells were harvested and analyzed at the indicated time points after infections.

**Preparation of lysates and subcellular fractions for RNA-seq and Ribo-seq**. 10$^7$ HEK 293T cells were seeded per Poly-L-Lysine-pretreated T150 plate. On the next day, plates were washed twice with PBS and incubated with CHIKV at a MOI of 4 for 1 h at 37 °C in DMEM without FBS. Plates were shaken every 10 minutes. After incubation the virus was removed and pre-warmed DMEM + 10% FBS was added. 12 h after infection (percentage of infected cells with one round of infection reaches maximum) cells were lysed or fractionated and harvested. To prepare total lysates ice cold lysis buffer (10 mM Tris-HCl pH 7.4, 10 mM MgCl$_2$, 100 mM NaCl, 1% Triton, 2 mM DTT, 0.25% Deoxycholate, 100 μg/ml CHX) was added to the cells, cells were scraped, transferred to a new tube and snap-frozen. Subcellular fractionation was based on a sequential detergent extraction as described in Jagannathan et al.[50]. with some modifications. Briefly, cells were preincubated for 3 min in DMEM containing 0.1 mg/ml CHX, washed in PBS + CHX and put on ice. All the following steps were performed on ice and with ice-cold reagents. Cells were incubated 5 min with 2,5 ml permeabilization buffer (110 mM KOAc, 20 mM K-Hepes pH 7.2, 5 mM MgCl$_2$, 1 mM EGTA, 0.03% digitonin, 1 mM DTT and 0.1 mg/ml CHX). The cytosol-containing supernatant was collected and snap-frozen. Subsequently, cells were washed in 5 ml washing buffer (same as permeabilization buffer but with 0.002% digitonin) and incubated for 5 min with 2.5 ml lysis buffer (200 mM KOAc, 20 mM K-Hepes pH 7.2, 5 mM MgCl$_2$, 1 mM EGTA, 2% dodecylmaltoside, 1 mM DTT and 0.1 mg/ml CHX). The membrane-containing supernatant was collected and snap-frozen. For further processing

samples were centrifuged 5 min at 12,000 g and the supernatants transferred to tubes for subsequent analysis.

**RNA and protein analysis**. Total RNA was extracted from HEK 293T cells by phenol-chloropropane extraction. Briefly, lysates were incubated 10 min at 60 °C with proteinase K (NEB), phenol-chloropropane extracted and precipitated with ethanol. The pellet was resuspended in RNAse-free water and treated with 0.1 μl Turbo DNAse/1 μg RNA (ThermoScientific) for 30 min at 37 °C. RNA was analyzed by qPCR using qScript XLT One-Step RT-qPCR ToughMix (Quanta BioSciences). 18 ng of total RNA were reverse transcribed and amplified using gene specific primers and TaqMan probes for all analyzed genes. The following primer/probes were used:

TaqMan GAPDH FW primer gccccggtttctataaattg, TaqMan GAPDH REV primer gtcgaacaggaggagcagaga, TaqMan GAPDH probe ccgcagcctcccgcttcg

TaqMan Jun FW primer ccggagcgaacttttgca, TaqMan Jun REV primer gccgccttctggtctttacc, TaqMan Jun probe ttcctgcgtcttaggcttcctccacgg

TaqMan TUBB6 FW primer gacagtgtgcatttccctcaag, TaqMan TUBB6 REV primer gtccttagtcctttcagcctcaga, TaqMan TUBB6 probe ctccggaaagtcacctgccaccc

TaqMan CHIKV NSP1 FW primer aaccccgttcatgtacaatgc, TaqMan CHIKV NSP1 REV primer gtacctgctcatctgcccaatt, TaqMan CHIKV NSP1 probe cgggtgcctacccctcatactcgac

TaqMan SKIL FW primer aaactgaatgggatgggagatg, TaqMan SKIL REV primer atttgcatgaatgtccgttatca, TaqMan SKIL probe tggcagcccccagcga

TaqMan SMAD6 FW primer tccgccacctccctactct, TaqMan SMAD6 REV primer gacaatgtggaatcggacagatc, TaqMan SMAD6 probe tctcctcgcgacgagtacaagccac

TaqMan HSP90B1 FW primer ttctttttgggagagacttgttttg, TaqMan HSP90B1 REV primer tgacccataatcccacattttaca, TaqMan HSP90B1 probe cccctaatcccttctcccctgca

TaqMan TMEM59L FW primer gccgatttgtggccagaa, TaqMan TMEM59L REV primer acgcaggctgcttcacact, TaqMan TMEM59L probe ctccaagcccaatgccacccaaac

For western blot analysis 3.3 μl NuPAGE LDS Sample buffer (4x) (ThermoScientific) were added to 10 μl lysate of the subcellular fraction and denatured for 10 min at 70 °C and loaded onto a 10% polyacrylamide-SDS gel. Following electrophoresis, proteins were transferred to a nitrocellulose membrane (GE Healthcare) for 1.5 h at 100 V on ice. Membranes were blocked for 1 h at room temperature in 5% nonfat dry milk in TBS-Tween (0.1% Tween 20) and then incubated overnight at 4 °C with primary antibodies diluted 1:1000 except otherwise indicated. The following antibodies to the following proteins were used: KIAA1456 (diluted 1:500; ThermoFisher, PA5-70320 and PA5-77162), Grp94 (Cell Signaling Technology, Cat#2104 S), TRAP (Santa Cruz, Cat#sc-376875), Tubulin (Sigma-Aldrich, Cat#T9026, RRID: AB_477593), Vinculin (Sigma-Aldrich, Cat#V9131), SLC25A45 (Novus Biological, Cat #NBP2-30521), PRSS16 (Novus Biological, Cat #NBP1-91559), DRP2 (Novus Biological, Cat #NBP2-16229), PHF7 (ThermoFisher, Cat#TA505115), KSR2 (Abcam, Cat#ab72753) and CHIKV Nsp1 and Capsid (diluted 1:5000 and kindly provided by A. Merits). Membranes were washed 3x for 10 min with TBS-Tween and subsequently incubated for one hour with horseradish peroxidase- or fluorescence-conjugated secondary antibody diluted 1:10000 (GE Healthcare Cat#NA934V and Cat#NA931V; Li-cor Cat#926-32210 and Cat#925-68073) diluted in 5% milk TBS-Tween at room temperature. After washing 3 more times for 10 min with TBS-Tween horseradish peroxidase-conjugated antibodies were labeled with SuperSignal West Femto Maximum Sensitivity Substrate (ThermoScientific) and fluorescence-conjugated antibodies were detected using the Licor system ODYSSEY. Band quantification was achieved with means of ImageGauge and ODYSSEY software analysis.

**KIAA1456 siRNA knock-down and overexpression experiments**. $5 \times 10^5$ HEK 293T cells were seeded in a 6 well-plate coated with Poly-L-Lysine. The next day, cells were pretreated with OptiMEM (ThermoFisher) for 30 min and then transfected using lipofectamine (i) with the (Myc-DDK-tagged) KIAA1456-ORF plasmid (Origene, Cat#RC203720) for the KIAA1456 overexpression experiments or (ii) with the ON-TARGETplus Non-Targeting Pool siRNA (Cultek, Cat#RC77D-001810-10-05) or the ON-TARGETplus SMARTPool Human KIAA1456 siRNA (Cultek, Cat#RC77L-HUMAN-XX-0005) for the KIAA1456 silencing experiments. In the case of overexpression 24 h later and in the case of silencing 48 h later, cells were infected with either MOI 4 or MOI 0.000004 and harvested 12 h post-infection. Infection outcome was measured via qPCR, Western Blot and standard plaque assay.

**Polysome profiling**. Polysome profiling was performed from cytosolic and membrane fraction lysates. Lysates were thawed at 25 °C and cleared for 5 min at 12000 g. 4 A260 units of cytosolic fractions and 2-4 A260 units of membrane fraction were loaded onto linear 15 % −45 % sucrose gradients and centrifuged for 2 h at 273865 g at 4 °C in a Beckman SW41 rotor. Gradients were prepared by underlying 45% sucrose dissolved in polysome buffer 1.0 (20 mM Tris-HCl pH 7.5, 10 mM MgCl$_2$, 100 mM KCl, 0.1 mg/ml CHX) a 15% sucrose solution and mixed using a Gradient Master. Lysates were treated with 20 U RNAse I (Ambion) per A260 unit of extract for 5 min at 25 °C. Digestion was stopped through the addition of 10 U RNase Inhibitor SUPERase-In (Ambion) per 20 U of RNAse I. Treated extracts were loaded onto linear 10 % − 50 % sucrose gradients prepared in polysome buffer 2.0 (20 mM Tris-HCl pH 7.5, 10 mM MgCl2, 100 mM NH4Cl,

0.1 mg/ml CHX) and centrifuged for 3 h at 209,678 $g$ at 4 °C in a Beckman SW41 rotor. Monosome peaks were collected and subjected to hot phenol RNA extraction in the presence of 1% SDS.

**Preparation of Ribo-seq and RNA-seq libraries.** Ribosome profiling (Ribo-seq) was carried out as previously described[51–54]. In brief, RNA extracted from monosomes was size-selected on a 15% polyacrylamide gel, dephosphorylated, ligated to a universal miRNA cloning linker (Biolabs) and subjected to rRNA subtraction using the Ribo-Zero Gold rRNA Removal kit (Epicenter). The purified adapter ligated RNA was reverse transcribed using SuperScript III (Thermo-Scientific), size-selected on a second 10% polyacrylamide gel, circularized with CircLigase II (Epicenter) and amplified with the Phusion polymerase (NEB). Oligos used were the RT_NatProt primer (AGATCGGAAGAGCGTCGTGTAGGGAAA GAGTGTAGATCTCGGTGGTCGC8CACTCA8TTCAGACGTGTGCTCTTCC GATCTATTGATGGTGCCTACAG) for reverse transcription and the Nat_Prot_PCR_fwd primer (AATGATACGGCGACCACCGAGATCTACAC) and Illumina index primers (NEBNext) for the PCR. Samples were sequenced with single-end 50 bp reads on an Illumina HiSeq2500 to at least 50 million reads per sample. RNA-seq libraries were prepared from 2, 5 µg total RNA (Proteinase K and Turbo RNA treated, as described in the RNA and protein analysis section) using the TruSeq stranded mRNA library prep kit (Illumina) following manufacturer's instructions and sequenced with paired-end 50 bp reads on an Illumina HiSeq2500 until reaching minimum 50 million reads/sample.

**Quantitation of tRNA modifications by LC-MS/MS.** HEK 293T and Huh7 cells were seeded in polylysine coated T100 plates at a density of $5 \times 10^6$ and $2 \times 10^6$ cells per plate, respectively. 24 h after seeding, cells were transfected using lipofectamine with the corresponding plasmid or infected with CHIKV, DENV or HCV as described in the previous section viruses and infection. 20 µg of DNAse-treated total RNA were ran in a 15% TBE-UREA gel (Novex, ThermoFisher Scientific) for 60 min at 180 V. Gels were stained with 1:5000 SybrGold (ThermoFisher Scientific) and the tRNA bands were excised. tRNA was then extracted using ZR small RNA PAGE Recovery kit (Zymo Research) following the manufacturer's instructions. 1 µg of the obtained tRNA was digested using the Nucleoside Digestion Mix (New England Bio Labs) according to the manufacturer's instructions and further desalted using HyperSep SpinTip Microscale (Thermo Fisher Scientific). Samples were run in the LTQ-Orbitrap XL mass spectrometer (Thermo Fisher Scientific, San Jose, CA, USA) coupled to an EASY-nLC 1000 (Thermo Fisher Scientific (Proxeon), Odense, Denmark). Ribonucleosides were loaded directly onto the analytical column and were separated by reversed-phase chromatography using a 50 cm homemade column with an inner diameter of 75 µm, packed with 4 µm Hydro-RP 80 Å (Phenomenex cat # 04A-4375), as previously described[55]. Chromatographic gradients started at 95% buffer A and 5% buffer B with a flow rate of 300 nl/min for 5 min and gradually increased to 20% buffer B and 80% buffer A in 40 min. After each analysis, the column was washed for 10 min with 20% buffer A and 80% buffer B. Buffer A: 20 mM Ammonium Acetate pH 4.5. Buffer B: 95% ACN/5% 20 mM Ammonium Acetate pH 4.5.

The mass spectrometer was operated in positive ionization mode with nanospray voltage set at 2 kV and source temperature at 200 °C. Full MS scans were set at 1 micro scans with a resolution of 60,000 and a mass range of m/z 100-700 in the Orbitrap mass analyzer. A list of masses was defined for further fragmentation (Supplementary Table 1). Fragment ion spectra were produced via collision-induced dissociation (CID) at normalized collision energy of 35% and they were acquired in the ion trap mass analyzer. Isolation window was set to 2.0 m/z and activation time of 10 ms. All data were acquired with Xcalibur software v2.1. Serial dilutions from commercial pure ribonucleosides (1–2000 pg/µl, Carbosynth, Toronto Research Chemicals) were used to establish the linear range of quantification and the limit of detection of each compound. A mix of commercial ribonucleosides was injected before and after each batch of samples to assess instrument stability and to be used as an external standard to calibrate the retention time of each ribonucleoside.

**Sequence alignment and data analysis.** RNA-sequencing (RNA-seq) reads were mapped against a combined human (hg38) and CHIKV (KT449801.1) genome using TopHat2 (v2.1.0, settings:—max-multihits 1—b2-very-sensitive—no-coverage-search—library-type fr-firststrand)[56] and supplying an additional transcriptome index based on GENCODE v24[48]. TIN scores were calculated using RSeQC's tin.py using the same transcriptome version in BED format.

Since mammalian transcripts usually feature several different splice isoforms that can feature different ORFs, we determined the most prominent mRNA isoform for each gene present in our RNA-seq samples to allow a gene-based analysis. To do so, we removed read-through and NMD transcripts from the Appris principal isoforms annotation (http://ubio.bioinfo.cnio.es/data/appris/) for hg38[57] (based on GENCODE v24) and quantified all remaining transcripts using kallisto (v0.43.0)[58], selecting the protein coding isoform which on average was expressed highest after sleuth's (v0.29.0)[59] between-sample normalization of the kallisto TPM values. In case multiple transcripts showed equal expression, they were ranked according to the following criteria: 1. Appris classification scheme ('PRINCIPAL:1', 'PRINCIPAL:2', 'PRINCIPAL:3', 'PRINCIPAL:4', 'PRINCIPAL:5',

'ALTERNATIVE:1', 'ALTERNATIVE:2'), 2. lowest CCDS and 3. lowest ENST ID. RNA-seq reads were subsequently quantified using featureCounts (v1.5.1, settings: -s 2 -Q 1 -p)[60] using our custom HEK 293T transcriptome annotation. Moreover, we also created a custom bowtie transcriptome index for our HEK 293T cells for ribosome footprint mapping to coding sequences. Prior to mapping to this custom index, we removed sequencing adaptors and trimmed the Ribo-seq reads using cutadapt (v1.91, settings: -a CTGTAGGCACCATCAAT—quality-base 33 -q 28,28 -m 25 -M 35)[61]. All surviving Ribo-seq reads were next mapped to the combined human (hg38) and CHIKV genome and read quantified using GENCODE v24 as described above. RNA biotype distributions (Supplementary Fig. 16) were generated for all RNA-seq and Ribo-seq samples by aggregating featureCounts read quantifications (GENCODE v24) into distinct categories based on type (see Supplementary Data 6 for details). Distributions of the relative read mapping locations were assessed via RSeQC's read_distribution.py (Supplementary Fig. 17a). Once this quality assessment had been completed, the trimmed Ribo-seq reads were mapped an index of known tRNA, rRNA, snRNA, snoRNA and miRNA genes based on GENCODE v24 via bowtie (v1.2.1)[62] and only non-mapping reads were used for mapping to the custom transcriptome index (bowtie v1.2.1, default settings) and subsequent analyses.

Read assignment to A- and P-site as well as footprint counting was performed using riboWaltz (v1.1.1), selecting reads of 29–33 nt length and using 14–18 nt as corresponding P-site offsets across all samples[63]. Read length distributions (Supplementary Fig. 17b), RPF extremity metaprofiles (Supplementary Fig. 18–21) and phasing distributions (Supplementary Fig. 22, 23) were generated via riboWaltz as well.

Assessing non-RPF contaminations of the Ribo-seq libraries was done by 1) quantifying RNA-seq and Ribo-seq reads mapping to GENCODE v24 long non-coding RNAs (lncRNAs) using featureCounts as before and thereafter calculating the normalized ribosome footprints ($\log_{10}$ mean (Ribo-seq counts) / mean (RNA-seq counts), Supplementary Fig. 4b), and 2) extracting and stratifying riboWaltz P-site counts for different RPF lengths across transcript regions (Supplementary Figs. 5–8).

Once RNA-seq and Ribo-seq reads had been quantified, we directly compared the corresponding counts for each sample to ensure that the expected near linear correlation between RPF and mRNA counts could be observed (Supplementary Fig. 24). Lastly, we also assessed replicate quality by sample clustering and by performing a principal component analysis on the $\log_{10}$-transformed count matrix (with added pseudocount) for the RNA-seq (Supplementary Fig. 25a, b) or Ribo-seq (Supplementary Fig. 25c, d) samples, which led to the exclusion of several samples in either technique (Supplementary Fig. 26). Supplementary Data 7 contains an overview summary of the individual samples.

All genes with fewer than a total of 5 RPF and 10 RNA-seq reads in either ER or cytosol were discarded before continuing our quantitative differential translation analysis. In addition, since RNA-seq reads mapping to the CHIKV SP region can originate from both the full-length CHIKV RNA (NSP ORF) as well as the subgenomic RNA (SP ORF), we calculated the average coverage across the NSP-specific region to estimate how many read counts mapping to the SP region should originate from the full-length (=NSP) isoform, assuming a uniform sequencing coverage. Using these estimates we then re-assigned a discrete number of reads from the SP to the NSP ORF for each sample prior to the differential expression analysis. We initially calculated normalized ribosome footprints (nRF) by dividing the mean counts per million (CPM) of Ribo-seq reads by the mean CPM of the RNA-seq reads of each gene. A joint quantitative analysis of both RNA-seq and Ribo-seq counts was then performed using limma-voom (v3.32.2)[26], where changes in relative ribosome footprint occupancy caused by CHIKV infection were independently modeled for ER and cytosol via an interaction term (~sequencing_type×infection_status). Results were visualized using ggplot2[64], GO term enrichment analyses were performed using the gProfileR package (v0.6.4)[65], while codon adaptation indices for each of the transcripts were calculated via the "cai" function from the seqinR package (v3.4–5)[66] using weights based on codon frequencies from CAIcal[67]. Codon frequencies were calculated by counting the occurrences of each of the 64 codon in an ORF / group of ORFs, and then dividing by the total number of codons of the ORF / group of ORFs. Different groups of ORFs were then compared by calculating the difference between their codon frequencies. Codon modification enzymes and their corresponding modifications (Supplementary Data 4) were based on ([28,68]), whereas codon statistics for other viruses (Supplementary Data 5) were based upon the codon usage tables provided by the Hive lab[69].

Mass spectrometry data were analyzed with the Skyline software (v20.2.0.343) using extracted precursor areas of the ribonucleosides for quantification.

**Reporting summary.** Further information on research design is available in the Nature Research Reporting Summary linked to this article.

## Data availability

Source data are provided as Source Data file. The Ribo-seq and RNA-seq data from this study have been submitted to the NCBI Gene Expression Omnibus (GEO) under accession number GSE143390. The raw proteomics data have been deposited to the MetaboLights study[70] repository with the dataset identifier MTBLS2443. Data resources

used in this study are: hg38 genome FASTA (https://hgdownload.soe.ucsc.edu/goldenPath/hg38/bigZips/hg38.fa.gz), CHIKV genome FASTA & annotation (https://www.ncbi.nlm.nih.gov/nuccore/KT449801.1), codon usage tables for different species (https://dnahive.fda.gov/dna.cgi?cmd=codon_usage&id=537&mode=cocoputs) and Appris principal isoforms (https://appris.bioinfo.cnio.es/#/downloads). Source data are provided with this paper.

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

## Acknowledgements

This work was supported by the Spanish Ministry of Science and Innovation (PID2019-106959RB-I00/AEI/10.13039/501100011033 and PCIN-2016-106 to JD and PGC2018-098152-A-100 to EMN) and by an institutional "María de Maeztu" Programme for Units of Excellence in R&D (CEX2018-000792-M) and by the 2017 SGR 909 grant from the Secretaria d'Universitats i Recerca del Departament d'Economia i Coneixement de la Generalitat de Catalunya. RB was a recipient of a Juan de la Cierva fellowship. Mass spectrometric analyses were performed in the CRG/UPF Proteomics Unit (Proteored, PRB3, grant PT17/0019 PE I + D + i 2013-2016, ISCIII and ERDF). We thank C. V. Nicchitta and S. Leidel for experimental advice and F. Gebauer and A. Meyerhans for fruitful discussions. We acknowledge the support of the MEIC to the EMBL partnership, Centro de Excelencia Severo Ochoa and CERCA Programme/Generalitat de Catalunya.

## Author contributions

J.J., R.B. and J.D. conceived the project. A.M., established the CHIKV system. J.J., M.T. and G.P.V. designed and executed the experiments. R.B. carried out the NGS data processing and analyses. R.B. and E.M.N. carried out the codon data analysis. J.J., R.B. and J.D. evaluated the results. J.J., R.B. and J.D. wrote the manuscript. All authors revised and commented on the manuscript.

## Competing interests

The authors declare no competing interests.
