## [Peer Review File · Nature Communications]

Reviewer comments, first round –

Reviewer #1 (Remarks to the Author):

Jungfleisch et al present here a very interesting work, in which they show that CHIKV infection causes an important remodelling of the translation landscape affecting mostly to mRNA associated to endoplasmic reticulum (ER), including the viral RNA. They observed that the codon optimality switches in infected cells and argue that for this to happen, it is required the activity of KIAA1456. While exciting, the study is still preliminary (correlative/observational) and lacks mechanistic insights. Below you will find my suggestions:

MAJOR POINTS

1. CHIKV, as other alphaviruses, induces a highly cytopathic infection, and the kinetic of infection in the experimental conditions used here and the timing of the experiments should be clearly elucidated/described. For example, authors use 40 MOI, which is known to cause an accelerated infection in Alphaviruses, making shorter the early phase where translation of the genomic RNA coexists with cellular protein synthesis. While I think this MOI is unnecessarily high, it can be acceptable if the timing of the experiment is fully disclosed in all figures and provide a comprehensive kinetic showing protein synthesis (S35 or fluorescent-puromycin labelling, RT-PCR and WB of viral proteins is provided). Knowing what the protein synthesis status is at the point of analysis is critical to interpretate the results. Alphaviruses cause a profound shutoff of protein synthesis and the translational landscape at the moment of the infection should be elucidated to contextualise these results.
2. In general terms, RNAseq data (ribosome profiling and transcriptome) should be quality controlled in the supplementary material, showing library depth, inter-replicate correlation, RNA biotype distribution, proportion of intron/exon mapping read (ribosome profiling is expected to be depleted of introns when compared to transcriptome), ribosome footprint density across the transcripts (5' UTR, ORF, 3' UTR). Etc, these general controls are important to evaluate the quality of the datasets. These should be applied to infected and uninfected cells separately to test for potential differences. Ribosome density should be plotted with and without normalisation to the transcriptome because the reasons listed below.
3. Ribosome profiling does not reveal translation perse but ribosomal footprints on the mRNA. places where the ribosomes are sitting. Unfortunately, there is not a clear way of differentiating active elongating ribosomes from 'frozen' ones and this should be clear in the text and properly balanced. Based on the translation efficiency estimation, translation of cytoplasmic mRNAs is not altered by the infection. However, we know that this is not the case, as S35 labelling is expected to show a massive shutoff of translation for most RNAs. This conundrum has a simple explanation, which is that authors report a substantial loss of cellular RNAs upon infection. Therefore, the normalisation of ribosome occupancy to RNA levels will show artificial translation efficiencies. This is why it is critical to, first, talk about normalised ribosomal footprints instead translation efficiencies

and, second, provide a direct comparison of ribosome occupancy vs RNA levels in infected and uninfected cells (e.g. scatter plot) using non-normalised data. These analyses can be expanded to cytosolic and ER transcripts separately to determine how these two components are affected depending on location.

4. To test the effects of differential ribosome occupancy in cytosol and ER in protein synthesis, orthogonal methods can be used to interrogate translation in a more direct way. For example, Western blotting against proteins whose cognate mRNA is regulated in either direction (or not regulated) in cytosol and ER. Alternatively, S35 labelling followed by immunoprecipitation and autoradiography can be used. Other possibilities will also be welcome.

5. Authors show that there is a loss of cellular RNAs that they relate to transcription (although high abundant and stable RNAs such as ACTB mRNA would not be affected by transcriptional regulation in a short timeframe). A recent work showed that in Sindbis virus this massive loss of cellular RNAs is due to RNA degradation and authors should consider this possibility as well. Said that, the loss of cellular RNAs will create artefacts when normalising the ribosome footprints to mRNA levels. If authors want to say that cytoplasmic translation is not affected, they should provide evidence with a method that measures translation directly or rather refer to ribosome occupancy.

6. Lane 109-113 - "The CHIKV RNAs and the translationally activated host mRNAs

differed in all examined features [length of CDS and untranslated regions, GC content, RNA secondary structure or codon usage bias] except in the codon usage bias". These results should be disclosed in the supplementary figures.

7. 4. FIG3 D. i) The effect of the overexpression of KIA1456 on mcm5 modification fig (Fig3 D) and on the viral infection (Fig3 E) is very intriguing. However, the phenotype is very mild and I suspect it is because the high multiplicity of infection. Experiments with lower multiplicity would likely increase the amplitude of the regulation. Moreover, authors focus on RNA levels, which is surprising as based on their working hypothesis the effects should be mostly at protein synthesis. Thus, other infection outcomes (viral proteins, viral particles in supernatant, etc) should be tested to complete the phenotypic profiling, at least, for the CHIKV experiment. ii) Moreover, complete the phenotype analysis, authors should show the effect of the depletion of KIA1456 (either with KD, KO) for both mcm5 modification and CHIV infection. ii) If possible to generate a catalytic mutant version of KIA1456, the effects of the overexpression of this mutant in KO or WT cells would provide further mechanistic insights. iii) Are the protein levels of virus-induced and plasmid KIAA1456 OE comparable? if not, why is this difference not visible in the methylation levels? (FIG S4) – add a WB panel showing the overexpression of KIA1456 in uninfected cells, with the same set (ctrl and OE) in infected cells in the same blot.

8. Does the overexpression of KIA increase the expression/translation of host mRNA with rare codons (for example some that are induced by CHIV)? This would prove the work model as correct.

MINOR POINTS

1. Authors show that there is a loss of cellular RNAs that they relate to transcription inhibition. However, in the short time frame of the infection, transcription regulation will mostly impact low abundant and unstable mRNAs. An alternative hypothesis is that RNA degradation is upregulated in virus infected cells and this has been shown recently for Sindbis virus, an Alphavirus closely related to CHIKV. This possibility should be discussed.

2. Authors use across the text the word 'expression' in a confusing way. They should clarify if referring to protein expression or RNA expression.

3. FIG 3C – In the WB with the time course analysis, can this analysis be quantified and analysed statistically?

4. FIG 3D – can the authors comment on the absence of decrease of mcm5

thiolation modification, since the enzymes involved in this reaction are downregulated after infection?

5. FIG S2 - Align the line for down and up regulated transcript to better mark the two different categories.

Reviewer #2 (Remarks to the Author):

In this manuscript, the authors explore how the Chikungunya virus expresses its genome at a high level despite an enrichment in rare codons. They analyzed translational changes of viral and cellular mRNAs upon CHIKV infection using a combined RNA-seq and ribosome profiling (Ribo-seq) approach. They showed that viral RNA translation and translational reprogramming occur mainly at the ER. This effect of CHIKV infection on translation is a result of reprogrammed codon usage, with the following five A-ending codons being among the most enriched in the translationally activated host and viral mRNAs: GAA, AAA, CAA, AGA and GGA. The decoding of these codons requires two modifications at the U34 position of relevant tRNAs, mcm5 and mcm5S2. These tRNAs appear to be methylated (mcm5) upon CHIKV infection through the methyltransferase activity of KIAA1456, a homolog of the TRM9 methyltransferase. They observe a 1.4-fold increase in KIAA1456 protein levels upon viral infection and overexpression of KIAA1456 facilitates CHIKV infection. The authors

further demonstrated that KIAA1456 also is induced by DENV, but not by HCV, consistent with enrichment in these same codons in DENV but not HCV. Overall, this is a very interesting finding that shows how viruses can alter the host epitranscriptome to favor viral production.

Major comments:

1. In general, some of the data are not yet convincing as there is no indication of number of repeats or statistics, data presentation is not clear and explanation of how statistical analyses were performed is missing.
2. In figure 3C and 3F, the authors show Western blots and claim that CHIKV and DENV infection, but not HCV infection, induces a moderate increase in KIAA1456 protein level. The Western blots are not convincing. They lack MW markers above and below the bands of interest and there is no indication of the number of repeats, how the quantification was performed or any statistical analyses. Why are there 2 bands in the DENV blot?
3. In Figures 3D, E and G, the bar graphs are not useful and it is difficult to evaluate the data as presented. Rather than bar graphs and SD or SE, the authors should explicitly show us all the data points in a scatter plot with the mean value indicated with a horizontal bar. Statistical analyses should be performed and a statement provided in the legend as to which statistical test was used and what the ** mean exactly. With the current presentation, it is impossible for the reader to evaluate the potential significance of these changes.
4. In figure 3D, the authors show a similar mcm5 increase (~1.3-fold) resulting from CHIKV infection and KIAA1456 mRNA overexpression. The authors claim that CHIKV induces a 1.4-fold increase in KIAA1456 protein levels (figure 3C), while overexpression of KIAA1456 resulted in a far more dramatic increase in the protein levels (figure S4). Why isn't the relative increase in mcm5 dose-dependent?
5. Given the relatively modest changes observed, in addition to overexpression of KIAA1456, the authors should perform a knockdown to show that the opposite result is obtained both on mcm5 levels and on viral RNA levels.
6. The statement on line 149 is too strong. The authors state: "This indicates that KIAA1456, like ALKHB8, catalyzes the required methylation to generate mcm5." In order to make this claim, the authors would have to purify the putative methyltransferase and demonstrate this activity in vitro. In the absence of this experiment, the claim should be toned down to "This suggests that.....".
7. Figure 2B: This graph took me a very long time to comprehend. Please fix the legend at the top as follows:

U34 modification:

Gray square unmodified (the tiny – is hard to see or understand)

Blue square mcm5/mcm5S2

Also, please define the equation used to determine the $\frac{mcm5}{mcm5S2}$ codon frequency. This is not clear. Is a 0.01 change significant?

8. Figure 2C and D: At first glance it is not clear why the changes in 2C are not significant (all dots are gray. It would be clearer to show data in 2C and 2D using the same axes scale and to point out in 2D

the dot corresponding to KIAA1456. Criteria for the assignment of the significance changes should be clearly indicated in the methods and legend.

9. The authors should provide an explanation for why the specific mRNAs were selected for validation in Figure S2.

10. Discussion: The authors do not mention anything about the Fig 2 results in the Discussion. Why were these results obtained and presented if they are not worth discussing? Is KIAA1456 translation localized to the ER for example? Also, the authors didn't observe an increase in mcm5S2 levels and the virus infection suppressed the 4 enzymes involved in thiolation. How is this consistent with decoding of the 5 A-ending codons? Is the thiolation not needed for some reason? Some discussion of why this makes sense in terms of the overall proposed mechanism is needed.

Minor comments:

1. Lines 90-92: the sentence "One group of mRNAs... displayed very low ones" is unclear. What does a "group" of mRNAs refer to?
2. Lines 136-141: it would be helpful to include more on the similarities/differences between the ALKHB8 and KIAA1456 homologs.
3. Line 169: KIAA1456 mistyped as KIAA146.
4. ER is not defined in the abstract.
5. Define KEGG and REACTOME (100), CDS (111), LC-MS/MS (145).
6. Figure 3 has 8 panels and suggest to divide into 4 separate figures (A, B, C-E, F+G)

Reviewer #3 (Remarks to the Author):

This well-written manuscript presents evidence for a very interesting story whereby the chikungunya virus suboptimal codon usage appears to be more optimal for translation than the standard cellular codon usage bias under conditions of virus infection, potentially as a result of increased KIAA1456 expression. The authors also show that the same may be true for a number of other viruses. While the authors' title "Virus-induced codon-specific reprogramming to favor viral RNA translation" tends to suggest the virus has evolved to reprogram the cellular translational machinery, perhaps a more accurate interpretation (as in the Discussion) may be that cell stress leads to reprogramming of codon optimality, and that viruses which cause cell stress have evolved their codon usage bias to reflect this endogenous reprogramming.

Weaknesses of the findings are that infection only led to a 1.4 fold increase in KIAA1456 protein expression levels (line 144) and overexpression of KIAA1456 led to only ~20% increase in CHIKV RNA during infection (Fig 3E). This is really a very small effect (normally in virus studies one would look for several-fold to several-log changes in virus load). On the other hand, this is exogenous overexpression on top of the endogenous overexpression as a result of virus infection, so maybe the true contribution of virus codon usage bias is masked.

Major comment:

The study depends heavily on ribosome profiling (Ribo-seq) data, yet does not show any quality control analyses to verify that the Ribo-seq data are clean and good quality. Contamination issues can be particularly problematic for infection data. See Irigoyen et al 2016 PMID 26919232 and supplementary material therein for discussion. At minimum, the authors should include as supplementary figures, for each library:

- Length distribution of host-CDS-mapping RPFs, with length distribution of virus-CDS-mapping RPFs from the same library overplotted.
- Phasing distribution of host-CDS-mapping RPFs and phasing distribution of virus-CDS-mapping RPFs.
- Metagene plots - i.e. mean RPF profile across host mRNAs, relative to start codons and relative to stop codons - showing near absence of RPFs in 3'UTRs and (hopefully) strong triplet phasing in CDSs.

The authors should also show Ribo-seq and RNA-seq mapped to the viral genome. Would it not be quite interesting to see if, for example, the large amount of SP cytoplasmic translation corresponds to RPFs in CP whereas when ribosomes reach the E3-E2-6K-E1 region maybe translation moves to the ER fraction?

Minor comments:

The authors should consider that not all viral RNA is being translated. Depending on the timepoint and protocol, there may be large amounts of genomic RNA that is destined for packaging and unavailable for translation. There can also be double-stranded genomic and subgenomic RNAs in replication complexes, again unavailable for translation. Thus it is actually not possible to determine viral TEs (although these effects would lead to TE underestimates, so do not negatively affect the conclusions). Maybe just add a bit of discussion text on this.

Sorry if I missed it, but I didn't see a clear statement of the number of Ribo-seq/RNA-seq biological repeats performed. E.g. in the Fig 2A caption, it just says "averaging the CPM per condition across all samples passing QC". It would be useful to have a Table summarizing the libraries (RNA-seq/Ribo-seq, infected/uninfected, timepoint, repeat number, number of host mRNA reads, number of virus reads).

Could the authors show some known ER host genes for non-infected libraries to show the Cyt/ER separation of RNA-seq/Ribo-seq is specific?

In Fig 1B, how was the threshold value 0.433 chosen?

Please show in supplementary the full gel images for the tiny WB figures (e.g. Figs 3F, 3C, S1 and S4).

In the figure captions please state the number of repeats and whether they are technical or biological (e.g. Figs 3D and 3E). In the text, please state p-values, fold changes, and type of statistical test used where significant changes are mentioned. Also I didn't see a definition for "***" in the figure captions.

Change "T"s to "U"s where referring to RNA/codons (e.g. Fig S3).

In Fig 3B, at the top next to "U34 modification", there is a grey box with a "-" to its right, which I assume means "no modification"? The "-" is very easy to miss so I think it would be clearer to write "no modification" or similar.

Relevant to Table S5, the authors might separate viruses according to chronic and acute viruses as the latter may tend to be more adapted to growth under conditions of cellular stress.

Do the authors have any thoughts as to why virus Cyt TE is much less than virus ER TE (Fig 2B)? Is this related to the KIAA1456 tRNA modification?

Line 63: "To investigate how CHIKV efficiently translates" -> "To investigate whether CHIKV efficiently translates".

Lines 69-70 - "RNA-seq analysis showed that CHIKV infection caused a massive decrease of cellular mRNA levels in both translation compartments": To be pedantic, there is no external normalization here - expression levels are normalized internally to the sum of virus- and host-mapping mRNA reads. Therefore it is theoretically possible that cellular mRNA levels are unchanged, but are just a

smaller fraction of the total due to the presence of virus reads in the infected samples. Therefore it is best to reword sentences like this to include e.g. "relative to total ...".

Lines 108 and 133: Change "codon usage" to "codon optimality" in these two subheadings. The codon usage is not changed just the relative optimality of these codons.

How does CHIKV codon usage bias compare to that of their mosquito hosts? Could that be a factor explaining their suboptimal codon usage relative to mammalian cells? Can the authors calculate the CAI of CHIKV relative to the codon usage of the translationally activated host genes (i.e. is CHIKV CAI low because CHIKV is adapted to the stress-conditions codon optimality or is it low because it is adapted to mosquito cells, or a mixture?).

What happens when a tRNA is left unmodified (mcm5 or mcm5s2)? Is it aminoacylated? Is it less efficient/slower/unable to function in translation?

Line 148: "significant increase in the levels of mcm5 modification (Fig 3D)" - list p-value, fold-change and test used. Similarly at line 153 for Fig 3E.

I think it would be useful to show a gene plot of KIAA1456 in supplementary, showing the RNA-seq and Ribo-seq reads mapped to the mRNA in uninfected and infected conditions.

For TE calculations, are the RNA-seq and Ribo-seq reads counted only within the CDS (else genes with longer UTRs will have lower TE estimates).

The authors haven't directly shown that KIAA1456 overexpression improves translational efficiency of the GAA, AAA, CAA, AGA and GGA codons. The authors might consider a reporter construct (e.g. luciferase) translated in wt or KIAA1456 overexpressing cells, one version of the reporter containing optimal Glu, Lys, Gln and Arg codons and another version containing GAA, AAA, CAA and AGA codons, and see if there are KIAA1456-specific changes in expression. (I can see there are some caveats with this approach though, so maybe not...).

REVIEWER COMMENTS:

Reviewer #1 (Remarks to the Author):

Jungfleisch et al present here a very interesting work, in which they show that CHIKV infection causes an important remodelling of the translation landscape affecting mostly to mRNA associated to endoplasmic reticulum (ER), including the viral RNA. They observed that the codon optimality switches in infected cells and argue that for this to happen, it is required the activity of KIAA1456. While exciting, the study is still preliminary (correlative/observational) and lacks mechanistic insights. Below you will find my suggestions:

MAJOR POINTS

1. CHIKV, as other alphaviruses, induces a highly cytopathic infection, and the kinetic of infection in the experimental conditions used here and the timing of the experiments should be clearly elucidated/described. For example, authors use 40 MOI, which is known to cause an accelerated infection in Alphaviruses, making shorter the early phase where translation of the genomic RNA coexists with cellular protein synthesis. While I think this MOI is unnecessarily high, it can be acceptable if the timing of the experiment is fully disclosed in all figures and provide a comprehensive kinetic showing protein synthesis (S35 or fluorescent-puromycin labelling, RT-PCR and WB of viral proteins is provided). Knowing what the protein synthesis status is at the point of analysis is critical to interpretate the results. Alphaviruses cause a profound shutoff of protein synthesis and the translational landscape at the moment of the infection should be elucidated to contextualize these results.

We used an MOI of 4 (titered in Hek293T) not of 40. We apologize for this typo. We have now corrected it and also included the corresponding viral kinetic showing viral RNA and protein expression and viral production (Fig. S1 a-c). All experiments were performed at 12 hours post infection, a time, at which there is a high viral production and no cytopathic effect. Full intracellular viral RNA and protein production were maintained at 24 hours post infection. At this time, cytopathic effects are observed in a small fraction of the cells, being complete at 36 hours post-infection.

2. In general terms, RNAseq data (ribosome profiling and transcriptome) should be quality controlled in the supplementary material, showing library depth, inter-replicate correlation, RNA biotype distribution, proportion of intron/exon mapping read (ribosome profiling is expected to be depleted of introns when compared to transcriptome), ribosome footprint density across the transcripts (5' UTR, ORF, 3' UTR). Etc, these general controls are important to evaluate the quality of the datasets. These should be applied to infected and uninfected cells

separately to test for potential differences. Ribosome density should be plotted with and without normalization to the transcriptome because the reasons listed below.

We have added the requested quality control plots (and some other additional ones) as Supplementary Figures S3, S5-9 and S11-S21 and provided an overview of the sequenced samples in Supplementary Tables S6-7.

3. Ribosome profiling does not reveal translation *per se* but ribosomal footprints on the mRNA. places where the ribosomes are sitting. Unfortunately, there is not a clear way of differentiating active elongating ribosomes from ‘frozen’ ones and this should be clear in the text and properly balanced. Based on the translation efficiency estimation, translation of cytoplasmic mRNAs is not altered by the infection. However, we know that this is not the case, as S35 labelling is expected to show a massive shutoff of translation for most RNAs. This conundrum has a simple explanation, which is that authors report a substantial loss of cellular RNAs upon infection. Therefore, the normalization of ribosome occupancy to RNA levels will show artificial translation efficiencies. This is why it is critical to, first, talk about normalized ribosomal footprints instead translation efficiencies and, second, provide a direct comparison of ribosome occupancy vs RNA levels in infected and uninfected cells (e.g. scatter plot) using non-normalized data. These analyses can be expanded to cytosolic and ER transcripts separately to determine how these two components are affected depending on location.

We thank the reviewer for the detailed comment. We are aware that paused ribosomes can artificially inflate the counts of ribosome protected fragments, however, we were unaware of any software that could deal with such events when we set out to analyze our data and developing such a tool from scratch was beyond the scope of this project. To accommodate the reviewer’s requests, we have changed the terminology in the manuscript to no longer use “translation efficiency” when referring to the Ribo-seq data, and instead introduced “normalized ribosome footprints” or make use of the common term “occupancy” throughout the text. Furthermore, we have included the requested scatter plot as Supplementary Figure S19.

As it has been demonstrated that the ER should be considered as a general site of cellular protein synthesis (reviewed in (Reid and Nicchitta, 2015)) and most mRNAs localize and translate both at the cytosol and the ER, sets of cytosolic and ER transcripts are not well defined. Hence, to expand the analysis to “cytosolic” and “ER” transcripts we determined based on our RNA-Seq data from uninfected cells transcripts to be cytosolic if they are at least twofold enriched in the cytosol versus the ER and vice versa for ER transcripts. The resulting scatter plots are included here in the response letter, as we did not observe any divergent behavior warranting their inclusion in the Supplement.

Please note that as shown in the Supplemental Fig S20 and S21, after assessing replicate quality by sample clustering and by performing a principal component analysis on the log₁₀-transformed count matrix, some of these samples were excluded for the analyses.

A) Cytosolic transcripts:

B) ER transcripts

4. To test the effects of differential ribosome occupancy in cytosol and ER in protein synthesis, orthogonal methods can be used to interrogate translation in a more direct way. For example, Western blotting against proteins whose cognate mRNA is regulated in either direction (or not regulated) in cytosol and ER. Alternatively, S35 labeling followed by immunoprecipitation and autoradiography can be used. Other possibilities will also be welcome.

We cannot carry out radiolabeled experiments in our S3 facilities. Thus, to be able to detect changes by Western blotting and circumvent protein stability issues, we focused on genes with higher absolute RPF numbers and for which there are antibodies available (Figure S4/ER translation).

5. Authors show that there is a loss of cellular RNAs that they relate to transcription (although high abundant and stable RNAs such as ACTB mRNA would not be affected by transcriptional regulation in a short timeframe). A recent work showed that in Sindbis virus this massive loss of cellular RNAs is due to RNA degradation and authors should consider this possibility as well. Said that, the loss of cellular RNAs will create artefacts when normalizing the ribosome footprints to mRNA levels. If authors want to say that cytoplasmic translation is not affected, they should provide evidence with a method that measures translation directly or rather refer to ribosome occupancy.

We thank the reviewer for pointing this out and modified the text accordingly. We added a sentence in page 3, lanes 77-78 indicating that RNA degradation might contribute to the massive decrease of mRNA levels.

We would also like to point out that our main differential analysis is detecting differences in Ribo-seq and RNA-seq counts before and after infection by accounting for the observed variance but does not perform a direct normalization of Ribo-seq to RNA-seq reads. In fact, such a normalization is used purely for the qualitative analysis shown in Fig. 2B. Nonetheless, as suggested in another comment, we are now referring to “normalized ribosome footprints” or “ribosome occupancy” instead of translation efficiency throughout the text.

Furthermore, we would like to clarify that we do not claim that cytosolic translation is unaffected, as we are aware of the fact that our dataset only offers a limited statistical power to detect such changes. To ensure that the readers do not misinterpret the corresponding results or our claims, we have added a remark to the corresponding results section on page 5, which now reads:

“After CHIKV infection, few significant translational changes were observed in the cytosol (Fig. 2C), which may be attributable to a limited statistical power in our dataset. Thus we focused our attention on the ER, where we observed dramatic changes (Fig. 2D) [...]”.

6. Lane 109-113 - “The CHIKV RNAs and the translationally activated host mRNAs differed in all examined features [length of CDS and untranslated regions, GC content, RNA secondary structure or codon usage bias] except in the codon usage bias”. These results should be disclosed in the supplementary figures.

We agree with the reviewer and have added the requested quality control plots as Supplementary Figures S5 and S6. Furthermore, we have modified this section of the results to provide a brief description of the findings. The section now reads:

“Overall, translationally activated host mRNAs tended to be similar to the CHIKV RNAs, featuring a longer CDS with a lower GC content when compared to host mRNAs that were either translationally repressed or not significantly altered (Fig. S5A-B). Likewise, these mRNAs were generally also less structured (Fig. S6A-B), although a comparison with CHIKV RNAs was not possible since the utilized Parallel Analysis of RNA Structure (PARS) scores were not available for the viral RNA.” (page 6)

7. FIG3 D (now Figure 5C). i) The effect of the overexpression of KIA1456 on mcm5 modification fig (Fig3 D, now Figure 5C) and on the viral infection (Fig3 E, now Figure 5D) is very intriguing. However, the phenotype is very mild and I suspect it is because the high multiplicity of infection. Experiments with lower multiplicity would likely increase the amplitude of the regulation. Moreover, authors focus on RNA levels, which is surprising as based on their working hypothesis the effects should be mostly at protein synthesis. Thus, other infection outcomes (viral proteins, viral particles in supernatant, etc) should be tested to complete the phenotypic profiling, at least, for the CHIKV experiment. ii) Moreover, complete the phenotype analysis, authors should show the effect of the depletion of KIA1456 (either with KD, KO) for both mcm5 modification and CHIV infection. iii) If possible, to generate a catalytic mutant version of KIA1456, the effects of the overexpression of this mutant in KO or WT cells would provide further mechanistic insights. iv) Are the protein levels of virus-induced and plasmid KIAA1456 OE comparable? if not, why is this difference not visible in the methylation levels? (FIG S4) – add a WB panel showing the overexpression of KIA1456 in uninfected cells, with the same set (ctrl and OE) in infected cells in the same blot.

Point i) and iv)

KIAA1456 protein expression levels achieved by plasmid-driven overexpression are 12-fold higher than the ones achieved via CHIKV infection. These results are now shown in Fig. 5B. As plasmid-driven KIAA1456 overexpression and CHIKV infection cause a similar increase in the levels of tRNA mcm⁵ modification, steps prior to the KIAA1456-catalyzed methylation might become limiting. One possible limiting step is the acetylation step directly prior to the methylation which is catalyzed by the acetyltransferase complex Elongator ELP1-6, a complex consisting of six subunits. In line with this, our ribosome profiling data show that the absolute RPF levels of all ELP subunits are downregulated after CHIKV infection. This comment is now included in page 9 (lanes 194-199).

Given the similar tRNA mcm5 modification levels observed under infection and overexpression conditions, the mild effect of KIAA1456 overexpression in CHIKV infection is not unexpected. At an MOI of 4 we observed a positive

effect in both viral RNA and protein levels (Fig. 5D-E). However, we did not detect any increment in the titer at this MOI or at lower MOIs (Fig. S10). This might be explained by the lower sensitivity of the plaque assay method in comparison with Western blot and qPCR methods. Alternatively, the amount of CHIKV RNA and protein produced in the infected cells might be in excess in relation to the production of infectious viral particles.

Point ii)

We tried different methods to KO/KD KIAA1456. We did not get any viable KO cell line and best silencing was obtained with siRNAs (around 50%, Fig. 5F). In agreement with a role of KIAA1456 in mcm⁵ tRNA modification and in CHIKV infection, KIAA1456 depletion caused a decrease in the level of mcm⁵ modification (Fig. 5G) and in viral RNA and protein levels (Fig. 5H-I). As in the KIAA1456 overexpression experiments, we did not observe any significant effect on the mean of the viral titers at MOI of 4 or lower. The lowest MOI we tested, 4×10^{-5} , was selected because at that MOI the virus (1 virus/well) has a 50% probability of infecting a cell with KIAA1456 silenced (take into account that the efficiency of siRNA transfection was around 50%). Thus, we expected a bimodal behavior with around 50% wells exhibiting reduced titers. Indeed, this is what we observed (Fig. S10).

Point iii)

To the best of our knowledge, no catalytic mutant version of KIAA1456 has been described. Methyl-transferase deficient ALKHB8 mutants have been generated by deletion studies (Begley, et al., 2013; Songe-Moeller et al., 2010). In contrast to its paralog ALKBH8, KIAA1456 seems to consist almost entirely of the S-adenosyl-L-methionine(SAM)-dependent methyltransferase domain, and hence thoroughly deletion studies must be carried to obtain a Methyl-transferase deficient mutant without affecting other protein properties. Although very interesting, we consider this to go beyond the scope of the manuscript.

8. Does the overexpression of KIA increase the expression/translation of host mRNA with rare codons (for example some that are induced by CHIKV)? This would prove the work model as correct.

We tested the effect of the overexpression of KIAA1456 on the validated genes in Figure S4 and we did not detect any effect (data not shown). However, one has to take into account that the effect of KIAA1456 under infection conditions occurs in the context of a massive mRNA degradation environment that ultimately is predicted to have profound alterations on the tRNA/mRNA ratios. It will be thus very interesting to study the effect of KIAA1456 overexpression in a

non-virus induced mRNA degradation environment. However, so far we did not find a suitable strategy to address it.

MINOR POINTS

1. Authors show that there is a loss of cellular RNAs that they relate to transcription inhibition. However, in the short time frame of the infection, transcription regulation will mostly impact low abundant and unstable mRNAs. An alternative hypothesis is that RNA degradation is upregulated in virus infected cells and this has been shown recently for Sindbis virus, an Alphavirus closely related to CHIKV. This possibility should be discussed.

We agree with the reviewer and have modified the text accordingly (page 3, lanes 78-79).

2. Authors use across the text the word 'expression' in a confusing way. They should clarify if referring to protein expression or RNA expression.

We apologize for the generated confusion and have modified the text accordingly.

3. FIG 3C (now Figure 5A) – In the WB with the time course analysis, can this analysis be quantified and analyzed statistically?

The quantification of the mean \pm SEM of KIAA1456 expression normalized to tubulin of n=3 independent samples is now indicated below the plot.

4. FIG 3D (now Figure 5C) - can the authors comment on the absence of decrease of mcm5 thiolation modification, since the enzymes involved in this reaction are downregulated after infection?

This is an intriguing matter. Indeed, a similar observation was obtained by overexpressing KIAA1456 in the highly tumorigenic cell line SW620 ((Begley et al., 2013), Supplemental table IV). Overexpression of KIAA1456 under translational stress conditions led to an increase of mcm⁵ but not of mcm⁵S². Our interpretation is that overexpression of KIAA1456 results in mcm⁵ modification of previously unmodified tRNAs resulting in an increase of the percentage of mcm⁵-modified tRNAs. If the Urm1, CTU1/2 and NFS1 enzymes required for thiolation would be limiting, the percentage of mcm⁵S²-modified tRNAs when comparing infected and non-infected cells would be conserved.

This is now discussed in the discussion section pages 10-11.

5. FIG S2 - Align the line for down and up regulated transcript to better mark the two different categories.

We aligned the lines.

Reviewer #2 (Remarks to the Author):

In this manuscript, the authors explore how the Chikungunya virus expresses its genome at a high level despite an enrichment in rare codons. They analyzed translational changes of viral and cellular mRNAs upon CHIKV infection using a combined RNA-seq and ribosome profiling (Ribo-seq) approach. They showed that viral RNA translation and translational reprogramming occur mainly at the ER. This effect of CHIKV infection on translation is a result of reprogrammed codon usage, with the following five A-ending codons being among the most enriched in the translationally activated host and viral mRNAs: GAA, AAA, CAA, AGA and GGA. The decoding of these codons requires two modifications at the U34 position of relevant tRNAs, mcm5 and mcm5S2. These tRNAs appear to be methylated (mcm5) upon CHIKV infection through the methyltransferase activity of KIAA1456, a homolog of the TRM9 methyltransferase. They observe a 1.4-fold increase in KIAA1456 protein levels upon viral infection and overexpression of KIAA1456 facilitates CHIKV infection. The authors further demonstrated that KIAA1456 also is induced by DENV, but not by HCV, consistent with enrichment in these same codons in DENV but not HCV. Overall, this is a very interesting finding that shows how viruses can alter the host epitranscriptome to favor viral production.

Major comments:

1. In general, some of the data are not yet convincing as there is no indication of number of repeats or statistics, data presentation is not clear and explanation of how statistical analyses were performed is missing.

We apologize for the missing descriptions. We have included now in the figure legends the number of independent replicates and how the statistical analyses were carried out. Furthermore, we exchanged all bar plots for dot blots to clearly present the single samples. All our data were achieved from at least 3 independent replicates. In addition, we have added a supplementary table (Table S7) containing details about the sequencing libraries (e.g. library depth, number of host and viral reads) and additional quality controls plots of the bioinformatic analyses (S3, S5-9 and S11-S21).

2. In figure 3C and 3F (now Figure 5A and 6A), the authors show Western blots

and claim that CHIKV and DENV infection, but not HCV infection, induces a moderate increase in KIAA1456 protein level. The Western blots are not convincing. They lack MW markers above and below the bands of interest and there is no indication of the number of repeats, how the quantification was performed or any statistical analyses. Why are there 2 bands in the DENV blot?

We apologize for the unclearness. The values were obtained from three biological replicates and quantifications were carried out using ImageGauge as stated in the material and method section. This information is now included in the figure legends. Furthermore, we included in the figures the MW marker bands and the average and SEM values. The KIAA1456 antibody that we used (ThermoFisher, PA5-70320) detects several bands besides the band corresponding to KIAA1456. We tested now additional KIAA1456 antibodies and selected one (ThermoFisher, PA5-77162) that is more specific. New Western blots were carried out and included now in Figure 6A.

3. In Figures 3D, E and G (now Figure 5C, 5D and 6B), the bar graphs are not useful and it is difficult to evaluate the data as presented. Rather than bar graphs and SD or SE, the authors should explicitly show us all the data points in a scatter plot with the mean value indicated with a horizontal bar. Statistical analyses should be performed and a statement provided in the legend as to which statistical test was used and what the * mean exactly. With the current presentation, it is impossible for the reader to evaluate the potential significance of these changes.

We exchanged the bar graphs for plots representing all data points and the mean \pm SEM. Furthermore, we explained in the figure legends that we performed two-tailed t-tests and changed ** by * (p-value lower than 0.05). Furthermore, we included in the manuscript the fold changes, p-values and statistical tests performed.

4. In figure 3D (now Figure 5C), the authors show a similar mcm5 increase (~1.3-fold) resulting from CHIKV infection and KIAA1456 mRNA overexpression. The authors claim that CHIKV induces a 1.4-fold increase in KIAA1456 protein levels (figure 3C), while overexpression of KIAA1456 resulted in a far more dramatic increase in the protein levels (figure 5B). Why isn't the relative increase in mcm5 dose-dependent?

To directly compare KIAA1456 expression levels upon CHIKV infection and KIAA1456 overexpression conditions we have added a new WB figure (Figure 5B) combining all samples. KIAA1456 protein expression levels achieved by plasmid-driven overexpression are 12-fold higher than the ones achieved via CHIKV infection. As plasmid-driven KIAA1456 overexpression and CHIKV infection cause a similar increase in the levels of tRNA mcm⁵ modification,

steps prior to the KIAA1456-catalyzed methylation might become limiting. One possible limiting step is the acetylation step directly prior to the methylation which is catalyzed by the acetyltransferase complex Elongator ELP1-6, a complex consisting of six subunits. In line with this, our ribosome profiling data show that the absolute RPF levels of all ELP subunits are downregulated after CHIKV infection. We have now included a paragraph in the results section (page 8-9, lanes 191-199) addressing this important point.

5. Given the relatively modest changes observed, in addition to overexpression of KIAA1456, the authors should perform a knockdown to show that the opposite result is obtained both on mcm5 levels and on viral RNA levels.

We tried different methods to KO/KD KIAA1456. We did not get any viable KO cell line and best silencing was obtained with siRNAs (around 50%, Fig. 5F). In agreement with a role of KIAA1456 in mcm⁵ tRNA modification and in CHIKV infection, KIAA1456 depletion caused a decrease in the level of mcm⁵ modification (Fig. 5G) and in viral RNA and protein levels (Fig. 5H-I).

6. The statement on line 149 is too strong. The authors state: "This indicates that KIAA1456, like ALKHB8, catalyzes the required methylation to generate mcm5." In order to make this claim, the authors would have to purify the putative methyltransferase and demonstrate this activity in vitro. In the absence of this experiment, the claim should be toned down to "This suggests that.....".

We followed the reviewer's suggestion (page 8, lane 187).

7. Figure 2B: This graph took me a very long time to comprehend. Please fix the legend at the top as follows:
U34 modification:
Gray square unmodified (the tiny – is hard to see or understand)
Blue square mcm5/mcm5S2
Also, please define the equation used to determine the \otimes codon frequency. This is not clear. Is a 0.01 change significant?

We thank the reviewer for pointing this out. To provide a clearer picture of the changes, we have introduced the following changes: (1) an additional Supplementary Figure (Fig. S8) which displays the frequency distributions in the translationally activated and repressed host mRNAs for each codon as well as the mean differences between the two distributions (mean(activated) – mean(repressed)). Here, all but one comparison would be highly significant ($p < 0.00001$) due to the high number of RNAs in each group (>3000). Thus, we chose to omit p-values. (2) Based on these distributions, directly use the mean differences in the bar plots (now moved to Fig. 4). We think that this approach is intuitive and still clearly displays all relevant results. Furthermore, as pointed out

by the reviewer, for clarity in Fig. 4 we now refer to U34 mcm5/mcm5S2 modification and “not modified” (gray square) and “modified” (blue square).

8. Figure 2C and D: At first glance it is not clear why the changes in 2C are not significant (all dots are gray. It would be clearer to show data in 2C and 2D using the same axes scale and to point out in 2D the dot corresponding to KIAA1456. Criteria for the assignment of the significance changes should be clearly indicated in the methods and legend.

Based on the reviewer’s suggestion, we added a red dot for KIAA1456 and ensured that Figure 2C and 2D use the same axis scales. As for the fact that all dots in Fig. 2C are grey, this is caused by a lack in statistical power due to having to remove low quality replicates. Since we have re-analyzed our data using read that map to CDS only (as suggested by another reviewer to rule out any biases caused by differences in UTR lengths), we are now able to detect a small number of significant changes, albeit the statistical power is still limited. As described in the Methods (page 23), significance is assigned by limma-voom when modelling our data using an interaction term between the type of the sequencing used (Ribo-seq or RNA-seq) and the status of the infection (infected or non-infected) (~ sequencing_type * infection_status).

To ensure that the readers do not misinterpret the corresponding results or our claims, we have added a remark to the corresponding results section on page 5, which now reads:

“After CHIKV infection, few significant translational changes were observed in the cytosol (Fig. 2C), which may be attributable to a limited statistical power in our dataset. Thus we focused our attention on the ER, where we observed dramatic changes (Fig. 2D) [...]”.

9. The authors should provide an explanation for why the specific mRNAs were selected for validation in Figure S2.

We selected mRNAs whose levels are significantly up- or downregulated in the Cyt and/or ER and belong to the group of mRNAs translationally activated or translationally repressed at the ER to cover all aspects (now included in the legend of Figure S2). In a final step, from these groups we chose those mRNAs for which we had primer/probes already validated in our lab

10. Discussion: The authors do not mention anything about the Fig 2 results in the Discussion. Why were these results obtained and presented if they are not worth discussing? Is KIAA1456 translation localized to the ER for example? Also, the authors didn’t observe an increase in mcm5S2 levels and the virus infection suppressed the 4 enzymes involved in thiolation. How is this consistent

with decoding of the 5 A-ending codons? Is the thiolation not needed for some reason? Some discussion of why this makes sense in terms of the overall proposed mechanism is needed.

We thank the reviewer for the comment. We have now included a paragraph describing a potential scenario explaining our data in the discussion (page 10-11, lanes 231-253).

Minor comments:

1. Lines 90-92: the sentence “One group of mRNAs... displayed very low ones” is unclear. What does a “group” of mRNAs refer to?

We referred to a “group of mRNA” on the basis of similar behaviors and similar TE values. To clarify this we now used the term “a subset” (lane 112).

2. Lines 136-141: it would be helpful to include more on the similarities/differences between the ALKHB8 and KIAA1456 homologs.

We thank the reviewer for pointing this out. We have now included further information about KIAA1456 and ALKHB8 in the manuscript (page 7-8, lanes 166-175).

3. Line 169: KIAA1456 mistyped as KIAA146.

We corrected this typo.

4. ER is not defined in the abstract.

We replaced the abbreviation with the complete word in the abstract.

5. Define KEGG and REACTOME (100), CDS (111), LC-MS/MS (145).

We followed the suggestion.

6. Figure 3 has 8 panels and suggest to divide into 4 separate figures (A, B, C-E, F+G)

We followed the suggestion.

Reviewer #3 (Remarks to the Author):

This well-written manuscript presents evidence for a very interesting story whereby the chikungunya virus suboptimal codon usage appears to be more optimal for translation than the standard cellular codon usage bias under conditions of virus infection, potentially as a result of increased KIAA1456 expression. The authors also show that the same may be true for a number of other viruses. While the authors' title "Virus-induced codon-specific reprogramming to favor viral RNA translation" tends to suggest the virus has evolved to reprogram the cellular translational machinery, perhaps a more accurate interpretation (as in the Discussion) may be that cell stress leads to reprogramming of codon optimality, and that viruses which cause cell stress have evolved their codon usage bias to reflect this endogenous reprogramming.

Weaknesses of the findings are that infection only led to a 1.4 fold increase in KIAA1456 protein expression levels (line 144) and overexpression of KIAA1456 led to only ~20% increase in CHIKV RNA during infection (Fig 3E). This is really a very small effect (normally in virus studies one would look for several-fold to several-log changes in virus load). On the other hand, this is exogenous overexpression on top of the endogenous overexpression as a result of virus infection, so maybe the true contribution of virus codon usage bias is masked.

Major comment:

1. The study depends heavily on ribosome profiling (Ribo-seq) data, yet does not show any quality control analyses to verify that the Ribo-seq data are clean and good quality. Contamination issues can be particularly problematic for infection data. See Irigoyen et al 2016 PMID 26919232 and supplementary material therein for discussion. At minimum, the authors should include as supplementary figures, for each library:

- Length distribution of host-CDS-mapping RPFs, with length distribution of virus-CDS-mapping RPFs from the same library overplotted.
- Phasing distribution of host-CDS-mapping RPFs and phasing distribution of virus-CDS-mapping RPFs.
- Metagene plots - i.e. mean RPF profile across host mRNAs, relative to start codons and relative to stop codons - showing near absence of RPFs in 3'UTRs and (hopefully) strong triplet phasing in CDSs.

We have added the requested quality control plots (and some other additional ones) as Supplementary Fig. S3, S5-9 and S11-S21 and provided an overview of the sequenced samples in Supplementary Tables S6-7.

2. The authors should also show Ribo-seq and RNA-seq mapped to the viral genome. Would it not be quite interesting to see if, for example, the large amount of SP cytoplasmic translation corresponds to RPFs in CP whereas when ribosomes reach the E3-E2-6K-E1 region maybe translation moves to the ER fraction?

Since we quantified the two viral ORFs independently, we think that such a behavior should be visible from the corresponding read counts, which does not appear to be the case. However, as requested, we generated an IGV overview of the mappings of all samples to the viral genome and included this plot below for the reviewer's information. As we did not notice any unexpected behavior and have already included a large number of Supplementary Figures, we believe it best not to include this image in the supplement. Note that in order to be able to visualize the vastly different read coverage for both infected and non-infected samples (see Table S7 for an overview of viral and host read counts), we had to enable log-scaling as well as auto-scaling.

Minor comments:

3. The authors should consider that not all viral RNA is being translated. Depending on the timepoint and protocol, there may be large amounts of genomic RNA that is destined for packaging and unavailable for translation. There can also be double-stranded genomic and subgenomic RNAs in replication complexes, again unavailable for translation. Thus it is actually not possible to determine viral TEs (although these effects would lead to TE underestimates, so do not negatively affect the conclusions). Maybe just add a bit of discussion text on this.

We thank the reviewer for pointing this out. The CHIKV (+)RNA contains two ORFs. The first ORF encodes the non-structural (NS) proteins and is expressed from the genomic RNA. The second ORF encodes the structural (SP) proteins and is expressed from a subgenomic RNA transcribed from a (-) RNA replication intermediate during infection. The genomic RNA has three functions, as mRNA for expression of the NSP ORF, as template for the synthesis of the (-) RNA replication intermediate and as template for encapsidation. In contrast, the subgenomic RNA mainly acts as mRNA for the synthesis of the SP ORF. As both CHIKV ORFs showed very similar behavior in our analysis any potential biases due to underrating the viral RNA available for translation seems negligible for our conclusions. In any case, we cannot completely eliminate the possibility that the observed effect in the genomic RNA is underestimated. We have included this point in page 4 (lanes 92-101).

4. Sorry if I missed it, but I didn't see a clear statement of the number of Ribo-seq/RNA-seq biological repeats performed. E.g. in the Fig 2A caption, it just says "averaging the CPM per condition across all samples passing QC". It would be useful to have a Table summarizing the libraries (RNA-seq/Ribo-seq, infected/uninfected, timepoint, repeat number, number of host mRNA reads, number of virus reads).

We apologize for the oversight and have included the missing table as Table S7.

5. Could the authors show some known ER host genes for non-infected libraries to show the Cyt/ER separation of RNA-seq/Ribo-seq is specific?

The supplemental Fig. S1D includes a validation of ER/Cyt separation via detection of known ER and cytosol markers by Western blot.

6. In Fig 1B, how was the threshold value 0.433 chosen?

The threshold was chosen arbitrarily based on previous work done in the lab during which it was decided that a 35% change in either direction should be considered as interesting ($\log_2(1.35) = 0.433$). We have added a note regarding this value to the figure legend.

7. Please show in supplementary the full gel images for the tiny WB figures (e.g. Figs 3F, 3C, S1 and S4).

This was also requested by the journal in an independent file which we already uploaded ("uncropped WB").

8. In the figure captions please state the number of repeats and whether they

are technical or biological (e.g. Figs 3D and 3E, now Figure 5C and D). In the text, please state p-values, fold changes, and type of statistical test used where significant changes are mentioned. Also I didn't see a definition for "***" in the figure captions.

We apologize for the unclearness. To easily evaluate the data we replaced bar plots with plots representing all data points (biological repeats). Furthermore, we have included in the figure legend the number of replicates, the definition of "***" (** was exchanged by *), the fold changes, p-values and statistical tests performed.

9. Change "T"s to "U"s when referring to RNA/codons (e.g. Fig S3).

We followed the reviewer's suggestion.

10. In Fig 3B, at the top next to "U34 modification", there is a grey box with a "-" to its right, which I assume means "no modification"? The "-" is very easy to miss so I think it would be clearer to write "no modification" or similar.

We followed the reviewer's suggestion.

11. Relevant to Table S5, the authors might separate viruses according to chronic and acute viruses as the latter may tend to be more adapted to growth under conditions of cellular stress.

This is a very interesting point. All viruses listed in Table S5 cause acute infections except HCV. This is the norm in (+)RNA viruses. However, in further work, it will be of great interest to compare chronic and acute viral infections in other viral groups.

12. Do the authors have any thoughts as to why virus Cyt TE is much less than virus ER TE (Fig 2B)? Is this related to the KIAA1456 tRNA modification?

This is an intriguing issue that we would like to address in the future. Currently, we are considering mainly two scenarios. One, because KIAA1456 is mainly translated at the ER upon infection, a gradient concentration might be generated with a higher KIAA1456 concentration around the ER. This would favor translation of the viral RNA, which is enriched in codons that are modified by KIAA1456, specifically at the ER. Two, some features of the ribosomes localized to the ER would favor translation of the viral RNA.

We have now included a paragraph about this in the discussion of the manuscript (page 10).

13. Line 63: "To investigate how CHIKV efficiently translates" -> "To investigate whether CHIKV efficiently translates".

We think that "how" is more accurate in this sentence as it is known that CHIKV RNA is very efficiently translated in infected cells

14. Lines 69-70 - "RNA-seq analysis showed that CHIKV infection caused a massive decrease of cellular mRNA levels in both translation compartments": To be pedantic, there is no external normalization here - expression levels are normalized internally to the sum of virus- and host-mapping mRNA reads. Therefore it is theoretically possible that cellular mRNA levels are unchanged, but are just a smaller fraction of the total due to the presense of virus reads in the infected samples. Therefore it is best to reword sentences like this to include e.g. "relative to total ...".

While it is true that we have not used any external normalization such as ERCC spike-ins, we have good reason to have chosen this wording as we have performed qPCR tests to see whether the viral reads replace a part of the host RNAs or are simply being added to the existing mix. This was done after finding that we were unable to independently validate the initial then DESeq2-based results for gene expression. Briefly, DESeq2's sizeFactors were artificially inflating the read counts to compensate for the drop in median expression caused by the sequencing coverage being spent on viral RNAs (see for instance this discussion on the Bioconductor forums: DESeq2- Size factors (median of ratios) assumptions - global adjustment normalization? (bioconductor.org)). After reanalyzing our data using limma-voom, we selected several genes that showed diverging behaviors in the two analyses and performed qPCR validations by correcting for the total input RNA, showing that host RNAs indeed decrease after infection (see Fig S2).

15. Lines 108 and 133: Change "codon usage" to "codon optimality" in these two subheadings. The codon usage is not changed just the relative optimality of these codons.

We followed the suggestion.

16. How does CHIKV codon usage bias compare to that of their mosquito hosts? Could that be a factor explaining their suboptimal codon usage relative to mammalian cells? Can the authors calculate the CAI of CHIKV relative to the codon usage of the translationally activated host genes (i.e. is CHIKV CAI low because CHIKV is adapted to the stress-conditions codon optimality or is it low because it is adapted to mosquito cells, or a mixture?).

The CAI values for CHIKV in human and in the two mosquito species *Aedes aegypti* and *Aedes albopictus* are 0.7107, 0.7412 and 0.7113, so there does not appear to be an adaptation to either host. Why this is the case and how the virus can replicate so well in such diverse host environments is part of an ongoing research project in our laboratory.

17. What happens when a tRNA is left unmodified (mcm5 or mcm5s2)? Is it aminoacylated? Is it less efficient/slower/unable to function in translation?

mcm⁵ and mcm⁵s² modifications are crucial to ensure translation efficiency and fidelity. Mechanistically, *in vitro* experiments demonstrate that both modifications enhance the affinity of the tRNA to its cognate codon in the ribosomal A-site and promote their translation by enhancing ribosomal A-site binding and dipeptide formation rates (Rezgui et al., 2013). Their roles have been mainly studied via the depletion of U₃₄-enzymes which leads to lower expression of mRNAs enriched in specific codons (AGA, AAA, GAA, CAA). E.g., loss of ELP3 in the cortex leads to codon-specific translation elongation defects with increased ribosome occupancy at codons read by U34-containing tRNAs, especially AGA (Laguesse et al., 2015). Here, the resulting inefficient decoding causes neurogenesis defects. Other recent studies link codon-dependent translation dysregulation to protein aggregation (Nedialkova and Leidel, 2015; Rapino et al., 2018). A very recent study shows a more complicated scenario as other mRNA features like hydrophilic sequences couple the defects in codon-dependent translation elongation to protein aggregation (Rapino et al., 2021).

18. Line 148: "significant increase in the levels of mcm5 modification (Fig 3D, now Figure 5C, lane 186)" - list p-value, fold-change and test used. Similarly at line 153 for Fig 3E (now Fig.5D, lane 200).

We followed the suggestions.

19. I think it would be useful to show a gene plot of KIAA1456 in supplementary, showing the RNA-seq and Ribo-seq reads mapped to the mRNA in uninfected and infected conditions.

We have added the requested plot as Figure S9.

20. For TE calculations, are the RNA-seq and Ribo-seq reads counted only within the CDS (else genes with longer UTRs will have lower TE estimates).

The reviewer raises a very good point. Our previous analyses (TE and differential ribosome occupancy) were based on RNA-seq counts from the entire mRNA, however, as the reviewer pointed out this could lead to a bias and

artificially separate mRNAs with different UTR lengths. We have therefore re-analyzed all our sequencing data and used reads mapping to CDS only to avoid such a bias. All affected Figures and Tables have been updated accordingly, and we are happy to report that our findings are robust and the conclusions remain unchanged.

21. The authors haven't directly shown that KIAA1456 overexpression improves translational efficiency of the GAA, AAA, CAA AGA and GGA codons. The authors might consider a reporter construct (e.g. luciferase) translated in wt or KIAA1456 overexpressing cells, one version of the reporter containing optimal Glu, Lys, Gln and Arg codons and another version containing GAA, AAA, CAA and AGA codons, and see if there are KIAA1456-specific changes in expression. (I can see there are some caveats with this approach though, so maybe not...).

We agree with the reviewer about the reporter approach. Unfortunately, how perturbations of codon-dependent translation impacts subsequent protein expression remains poorly understood. In a very recent study it was demonstrated that the mRNA codon content is not sufficient to predict protein abundance. Specifically, they show that translation defects due to the loss of U34-enzymes are not systematically associated with changes in protein expression. They propose a model in which hydrophilic amino acid patterns link translation defects to protein aggregation and degradation (Rapino et al., 2021). In this model, tRNA modification and maintenance of protein integrity are two independent processes. Due to this complexity and the still poorly understood coupling of codon-dependent translation rates and protein output we consider that we do not have enough knowledge yet to design and evaluate proper reporter constructs.

References:

- Begley, U., Sosa, M.S., Avivar-Valderas, A., Patil, A., Endres, L., Estrada, Y., Chan, C.T., Su, D., Dedon, P.C., Aguirre-Ghiso, J.A., *et al.* (2013). A human tRNA methyltransferase 9-like protein prevents tumour growth by regulating LIN9 and HIF1- α . *EMBO molecular medicine* 5, 366-383.
- Laguesse, S., Creppe, C., Nedialkova, D.D., Prevot, P.P., Borgs, L., Huysseune, S., Franco, B., Duysens, G., Krusy, N., Lee, G., *et al.* (2015). A Dynamic Unfolded Protein Response Contributes to the Control of Cortical Neurogenesis. *Developmental cell* 35, 553-567.
- Nedialkova, D.D., and Leidel, S.A. (2015). Optimization of Codon Translation Rates via tRNA Modifications Maintains Proteome Integrity. *Cell* 161, 1606-1618.
- Rapino, F., Delaunay, S., Rambow, F., Zhou, Z., Tharun, L., De Tullio, P., Sin, O., Shostak, K., Schmitz, S., Piepers, J., *et al.* (2018). Codon-specific translation reprogramming promotes resistance to targeted therapy. *Nature* 558, 605-609.
- Rapino, F., Zhou, Z., Roncero Sanchez, A.M., Joiret, M., Seca, C., El Hachem, N., Valenti, G., Latini, S., Shostak, K., Geris, L., *et al.* (2021). Wobble tRNA modification and hydrophilic amino acid patterns dictate protein fate. *Nature communications* 12, 2170.

Reid, D.W., and Nicchitta, C.V. (2015). Diversity and selectivity in mRNA translation on the endoplasmic reticulum. *Nature reviews Molecular cell biology* 16, 221-231.

Rezgui, V.A., Tyagi, K., Ranjan, N., Konevega, A.L., Mittelstaet, J., Rodnina, M.V., Peter, M., and Pedrioli, P.G. (2013). tRNA tKUUU, tQUUG, and tEUUC wobble position modifications fine-tune protein translation by promoting ribosome A-site binding. *Proceedings of the National Academy of Sciences of the United States of America* 110, 12289-12294.

L. Songe-Moller, E. van den Born, V. Leihne, C. B. Vagbo, T. Kristoffersen, H. E. Krokan, F. Kirpekar, P. O. Falnes, A. Klungland (2010). Mammalian ALKBH8 possesses tRNA methyltransferase activity required for the biogenesis of multiple wobble uridine modifications implicated in translational decoding. *Molecular and cellular biology* 30, 1814-1827.

Reviewer comments, second round –

Reviewer #1 (Remarks to the Author):

The authors have answered most of my comments and beyond very minor comments, there is no more additional experiments from my side.

1. I thank the authors for indicating that the MOI is 4, which aligns better with most studies done with alphaviruses. In figure S1A there is capsid signal in the - infection lanes. Are these spill overs of gel loading or true signal?

2. In non-normalised conditions, the reads mapping ORFs is much lower in infected cells. This agrees with the shut off of protein synthesis occurring in alphavirus infected cell, which correlates with a loss of cellular mRNA (here why normalisation closes the gap between the 2 conditions in figure S3B).

3. Removing outlier replicates with evident deficiencies in quality is ok always that this is shown with data and it is disclaimed in the text and figures as the authors did in the supplementary figures.

4. Fully answered

5. Overall shut off of protein synthesis in alphavirus infection have been largely shown by S35 labelling and polysome profiling (with a remarkably loss of methionine signal in cellular proteins in infected cells in the first and a loss of polysomes in the second). I am aware that S35 labelling and polysome profiling will be clearly biased to the most abundant mRNAs in the cell, so the beauty of the present data is that can provide information for the rest. Therefore, the authors' suggestion is fine always these critical information is clear in the text.

6. Good

7. It is nice to see that despite the low efficiency of the siRNA, effects are observed at the RNA level. The authors have indeed tried different approaches to answer the query. Just a minor point: as several of the experiments have a high variability, it would be good to see dots coloured by replicate. In this way, it would be possible to distinguish between an effect that is present across replicates but with different dynamic ranges or signal to noise ratios, from a low reproducibility observation that benefits from a limited number of replicates with high effects.

Reviewer #2 (Remarks to the Author):

The reviewers have responded thoroughly to all of my comments.

Reviewer #3 (Remarks to the Author):

I thank the authors for addressing my comments. I have two major comments on the revised manuscript plus a few minor ones.

MAJOR COMMENT 1:

In my earlier review I had noted that "The study depends heavily on ribosome profiling (Ribo-seq) data, yet does not show any quality control analyses to verify that the Ribo-seq data are clean and good quality. Contamination issues can be particularly problematic for infection data. See Irigoyen et al 2016 PMID 26919232 and supplementary material therein for discussion." I went on to

request a minimum set of quality control plots to show for each NGS library.

The authors have now included quality control analyses - thank you. However, these now highlight some serious issues with the RiboSeq data that, in my opinion, makes it unusable (unless - see below - there is just a 3'UTR annotation issue in the data processing pipeline).

For virus infection data, we and others have often found that, at late time points of infection, Ribo-seq data can become horribly contaminated by non-RPF-derived RNA. Depending on the virus species, this can affect virus reads, or host reads, or both. Possible factors include RNA degradation, virus RNA-binding proteins (e.g. capsid) binding to virus (and sometimes also host) RNA and cosedimenting with ribosomes, translational shutoff leading to a greatly increased contamination:RPF ratio simply because the true RPF quantities are greatly decreased, and potentially other unknown factors.

Contamination can be gauged either via triplet phasing or (in cases where triplet phasing is poor e.g. due to poor nuclease trimming) by a near-absence of Ribo-Seq reads mapping to 3'UTRs in host "metagene" plots. If the latter, and since virus and host can be subject differentially to contamination, assessment of contamination in virus could depend on quantifying differences (or showing that there are no differences) between the virus and host RPF length distributions.

Unfortunately, many of the authors' RiboSeq samples have very poor triplet phasing. Particularly noteworthy is that the triplet phasing in virus-mapping reads is much worse than in host-mapping reads (Fig S17-18) which shows that the virus Ribo-Seq is for sure highly contaminated. The manuscript depends mostly on host differential expression analysis so dumping the virus RiboSeq might not be catastrophic. However, from the host "metagene plots" (Fig S3, S13-16), we can see that we have a very high density of Ribo-seq reads mapping to the 3'UTRs. This is particularly bad in the infected/ER samples (Fig S3D - ER plot - orange line) where we can very little drop in read density at the termination codon (instead of an expected 90-99% sudden drop). There is a gradual drop off in density as we go further into the 3'UTR, but this is a known characteristic - typically also observed in the matched RNAseq (which the authors haven't shown here) and which maybe be a result of some transcripts in the physical sample having shorter UTRs than the corresponding gene annotations (we see a similar ramp up in the 5'UTR). The host triplet phasing is also not great - e.g. in Fig S15 (ER, infected samples), instead of seeing 90%+ of reads mapping to the first nucleotide of codons, we see much more similar %s of reads mapping to each of the 3 codon positions. Thus there is no evidence to exclude the possibility that - especially in the ER/infected samples - a large proportion of the reads may be (indeed are likely to be) coming from non-RPF sources. If it was me, I would not trust or use such data sets.

- Fig S3D, ER plot (i.e. lower panel), orange line (i.e. infected): The orange line in the 3'UTR region is as high (or even higher than) the orange line in some CDS regions, whereas it should be 90-99% lower than the CDS density. The 3'UTR density estimates the contamination level.

- Fig S13-S16: The mocks are not as bad as the infected samples, though the 3'UTR density still seems rather high. In the ER/infected samples (Fig S15), there is no qualitative difference in read density between the CDS and 3'UTR regions. There does seem to be some sort of triplet periodicity extending into the 3'UTRs - could the 3'UTR density be an artifact of misannotation (in which case the data may be OK after all!).

- Fig S17-18: Note how in the infected samples (*), in the CDS region (middle panels), some phasing is existent in host reads but phasing is almost indiscernible in the virus reads - suggesting that a very large proportion of the virus reads are contamination.

MAJOR COMMENT 2:

Alphaviruses (and also dengue virus but not HCV) are dual host viruses, replicating alternately in vertebrate hosts and mosquito hosts, which have different codon usage biases. Thus, one can imagine that failure of chikungunya virus to mimic optimal human codon usage could, at least in part, be driven by competing selection to mimic mosquito codon usage. The authors didn't fully

address my previous comment on this and, so far as I can see, there is also no mention of this issue in the manuscript. I feel that it is essential to at least mention this issue in the manuscript to give a proper background to understanding the possible reasons why CHIKV codon usage is suboptimal.

Moreover, the authors didn't address the part of my previous comment where I asked "Can the authors calculate the CAI of CHIKV relative to the codon usage of the translationally activated host genes (i.e. is CHIKV CAI low because CHIKV is adapted to the stress-conditions codon optimality or is it low because it is adapted to mosquito cells, or a mixture?)." In the supplementary table showing codon usage in human, CHIKV and various other viruses, could the authors calculate a codon usage table for the human set of translationally activated genes, and then using this table as the baseline, calculate the CAI for all the rows in this table, and add this as another column. It will be important to see whether the CAI for CHIKV is substantially increased when calculated relative to human translationally activated genes, and also how this compares with the CHIKV CAI relative to its mosquito host.

Maybe it would also be useful to add a panel to Fig 4, comparing CHIKV vs Aedes?

MINOR COMMENTS:

- In the "Uncropped western blots" document, page 6, lower blot - I am curious as to why the marker lane (and some adjacent lanes) appear to disappear above 49 kDa? Could the greyscale be adjusted so the composition of this blot is more clear?

- Are the data in Fig S3 averages over all libraries?

- In the Fig S3 caption, I found the meaning of "regions" unclear. The first sentence suggests the regions are 5'UTR, CDS and 3'UTR. However, this is not compatible with the later statement "... normalized by the total number of RPFs in the region", since in that case, in each normalized plot, the CDS mean value, 5'UTR mean value, and 3'UTR mean value should all be equal to each other. Instead it looks like the "region" for normalization is the full-length mRNA, so that full-length mRNA infected and full-length mRNA control have the same mean values.

- In Fig S3A and S3B, the x-axis scale is confusing and I did not see an explanation. Is it % of the way through the annotated 5'UTR, CDS or 3'UTR, respectively?

- In Fig S3, I assume it is P-site mapping, but this should be clarified in the caption.

- In Fig S13-S16, the caption says P-site positions are being plotted. Please check the location of "0" and the red vertical line on each plot. It is a bit unusual that the main jump up in density from 5'UTR to CDS, and jump down in density from CDS to 3'UTR seems to occur a little offset from the red line (expect it to be one codon upstream at the stop codon since termination occurs with the stop codon in the A-site).

- Lines 225-226: The authors say "Our study identifies an unanticipated virus-induced adaptation of the host translation machinery to the viral codon usage." I think it is wrong to say that the host has adapted to the viral codon usage. More likely that cell stress leads to reprogramming of codon optimality as a generic host response to deal with stress, and that viruses which cause cell stress have evolved their codon usage bias to reflect this endogenous reprogramming - i.e. the virus has adapted for optimal gene expression in the host infection environment.

REVIEWER COMMENTS

Reviewer #1 (Remarks to the Author):

The authors have answered most of my comments and beyond very minor comments, there is no more additional experiments from my side.

1. I thank the authors for indicating that the MOI is 4, which aligns better with most studies done with alphaviruses. In figure S1A there is capsid signal in the - infection lanes. Are these spill overs of gel loading or true signal.

Thank you for pointing this out. The signal comes from small spillovers that occurred during the loading of the gel. We have now substituted the Western by a new one (Fig S1A).

2. In non-normalised conditions, the reads mapping ORFs is much lower in infected cells. This agrees with the shut off of protein synthesis occurring in alphavirus infected cell, which correlates with a loss of cellular mRNA (here why normalization closes the gap between the 2 conditions in figure S3B).

Agreed

3. Removing outlier replicates with evident deficiencies in quality is ok always that this is shown with data and it is disclaimed in the text and figures as the authors did in the supplementary figures.

The removal of the outliers is disclaimed in the results section, explained in the method section and visualized in the supplementary Figures S25 and S26 (previously S20 and S21).

4. Fully answered

5. Overall shut off of protein synthesis in alphavirus infection have been largely shown by S35 labelling and polysome profiling (with a remarkably loss of methionine signal in cellular proteins in infected cells in the first and a loss of polysomes in the second). I am aware that S35 labelling and polysome profiling will be clearly biased to the most abundant mRNAs in the cell, so the beauty of the present data is that can provide information for the rest. Therefore, the authors' suggestion is fine always these critical information is clear in the text.

We have added an additional sentence pointing towards the shutoff we observe here upon CHIKV infection and which is known to occur upon alphavirus infection (page 5).

6. Good

7. It is nice to see that despite the low efficiency of the siRNA, effects are observed at the RNA level. The authors have indeed tried different approaches to answer the query. Just a minor point: as several of the

experiments have a high variability, it would be good to see dots coloured by replicate. In this way, it would be possible to distinguish between an effect that is present across replicates but with different dynamic ranges or signal to noise ratios, from a low reproducibility observation that benefits from a limited number of replicates with high effects.

Each dot represents an independent replicate.

Reviewer #2 (Remarks to the Author):

The reviewers have responded thoroughly to all of my comments.

Reviewer #3 (Remarks to the Author):

I thank the authors for addressing my comments. I have two major comments on the revised manuscript plus a few minor ones.

MAJOR COMMENT 1:

...For virus infection data, we and others have often found that, at late time points of infection, Ribo-seq data can become horribly contaminated by non-RPF-derived RNA. Depending on the virus species, this can affect virus reads, or host reads, or both. Possible factors include RNA degradation, virus RNA-binding proteins (e.g. capsid) binding to virus (and sometimes also host) RNA and cosedimenting with ribosomes, translational shutoff leading to a greatly increased contamination:RPF ratio simply because the true RPF quantities are greatly decreased, and potentially other unknown factors.

Contamination can be gauged either via triplet phasing or (in cases where triplet phasing is poor e.g. due to poor nuclease trimming) by a near-absence of Ribo-Seq reads mapping to 3'UTRs in host "metagene" plots. If the latter, and since virus and host can be subject differentially to contamination, assessment of contamination in virus could depend on quantifying differences (or showing that there are no differences) between the virus and host RPF length distributions.

Unfortunately, many of the authors' RiboSeq samples have very poor triplet phasing. Particularly noteworthy is that the triplet phasing in virus-mapping reads is much worse than in host-mapping reads (Fig S17-18) which shows that the virus Ribo-Seq is for sure highly contaminated. The manuscript depends mostly on host differential expression analysis so dumping the virus RiboSeq might not be catastrophic.

However, from the host "metagene plots" (Fig S3, S13-16), we can see that we have a very high density of Ribo-seq reads mapping to the 3'UTRs. This is particularly bad in the infected/ER samples (Fig S3D - ER plot - orange line) where we can very little drop in read density at the termination codon (instead of an expected 90-99% sudden drop). There is a gradual drop off in density as we go further into the 3'UTR, but this is a known characteristic - typically also observed in the matched RNAseq (which the authors haven't shown here) and which maybe be a result of some transcripts in the physical sample having shorter UTRs than the corresponding gene annotations (we see a similar ramp up in the 5'UTR). The host triplet phasing is also not great - e.g. in Fig S15 (ER, infected samples), instead of seeing 90%+ of reads mapping to the first nucleotide of codons, we see much more similar %s of reads mapping to each of the 3 codon

positions. Thus there is no evidence to exclude the possibility that - especially in the ER/infected samples - a large proportion of the reads may be (indeed are likely to be) coming from non-RPF sources. If it was me, I would not trust or use such data sets.

- Fig S3D, ER plot (i.e. lower panel), orange line (i.e. infected): The orange line in the 3'UTR region is as high (or even higher than) the orange line in some CDS regions, whereas it should be 90-99% lower than the CDS density. The 3'UTR density estimates the contamination level.
- Fig S13-S16: The mocks are not as bad as the infected samples, though the 3'UTR density still seems rather high. In the ER/infected samples (Fig S15), there is no qualitative difference in read density between the CDS and 3'UTR regions. There does seem to be some sort of triplet periodicity extending into the 3'UTRs - could the 3'UTR density be an artifact of misannotation (in which case the data may be OK after all!).
- Fig S17-18: Note how in the infected samples (*), in the CDS region (middle panels), some phasing is existent in host reads but phasing is almost indiscernible in the virus reads - suggesting that a very large porportion of the virus reads are contamination.

We thank the reviewer for these thorough comments. We deliberately chose an advanced time of infection (high viral RNA levels, high viral expression but still viable cells) for the analyses because we reasoned that at this time point, any modification of the translation machinery should be easier to detect. Indeed, because of this decision, we were able to uncover that cellular stress caused by viral infections, which is not present at early time points, leads to a reprogramming of codon optimality and that viruses have evolved their codon usage bias to adapt to it. However, we understand that such an advanced time of infection intrinsically hampers ribosome profiling analyses, especially when combined with a subcellular fractionation of the infected cells.

Related to the triple periodicity in virus-mapping reads (former Fig S17-18, now Fig. S22-23), we agree that the virus Ribo-seq analyses may suffer from contaminations due to the chosen time of the analyses as exemplified by the absence of a clear triplet periodicity signal. Such contaminants can result in an overestimation of the actual ribosome occupancy, which is why we have included a corresponding statement in the results section (page 5-6) to clarify this fact to the reader. However, as the reviewer points out, our subsequent analyses focus on the behavior of host transcripts.

In order to address the reviewer's concerns over signal originating from the 3'UTRs of transcripts after infection, we included several additional quality controls. First, we assessed the level of RNA degradation due to viral infection and host transcription shut-off by calculating the transcript integrity number (TIN) for each transcript based on our RNA-seq data. The results are shown in the new Figure S4A and confirm a significant decrease of the average TIN for ER samples after infection, which may indeed negatively affect our estimates. In addition, we also re-analyzed our Ribo-seq data using the longest ORF for each gene, however, this did not have any impact on triplet periodicity or metagene plots, which is why we chose not to include these results here. Next, we jointly investigated the dominant reading frame and RPF length distributions across the CDS and UTRs, which have been included as new Figures S5-S8. Based on these new Figures we make two observations: 1) no triplet periodicity indicative of stop-codon readthrough can be seen in the 3'UTRs, and 2) the RPF length distributions of CDS and 3' UTR are almost identical. In the case of random binding or co-sedimentation of RNA-binding proteins or other

contaminants, we would expect to see a marked difference in the length distributions. Moreover, we would also expect such events not to be limited to only one region of the transcript (such as the 3' UTR) and instead to occur in a random fashion across all mRNAs and mRNA-like transcripts. Nevertheless, this anomaly does not occur in the 5'UTRs. For this reason, we calculated the normalized ribosomal footprints for long non-coding RNAs before and after infection akin to Fig 2B (new Figure S4B). In the case of random binding or co-sedimentation of contaminants we expect to see a marked increase of false ribosome occupancy. While we do see such an increase for the cytosol, this is not the case for the ER. In combination with the previously mentioned absence of phasing and near identical RPF length distribution in the 3'UTR, these results lead us to conclude that the effect of such contamination on the host transcripts seems to be limited at best. Instead, the signal originating from the 3'UTR (see metagene plots in Figure S3) could be explained by virus-induced problems with the post-termination ribosome recycling leading to non-translating ribosomes remaining bound to the 3' UTR. In agreement with this, we noted that ABCE1, a key component in the ribosome recycling machinery, is down-regulated after CHIKV infection in both the ER and cytosol translation compartments (ER: RNA log₂FC = -6.56, RPF log₂FC = -3.21; cytosol: RNA log₂FC = -1.81, RPF log₂FC = -1.35).

To better explain the limitations of our analyses, we have included a new paragraph in the result section (pages 5-6). Nevertheless, despite these limitations, our ribosome profiling analyses guided us to hypothesize the importance of the KIAA1456 enzyme in CHIKV RNA translation. Moreover, (i) we validated the RNA-seq and Ribo-seq analyses in an independent experiment by quantitative PCR and Western blot, and importantly, (ii) we experimentally prove the role of KIAA1456 and the related tRNA modification(s) using multidisciplinary approaches.

MAJOR COMMENT 2:

Alphaviruses (and also dengue virus but not HCV) are dual host viruses, replicating alternately in vertebrate hosts and mosquito hosts, which have different codon usage biases. Thus, one can imagine that failure of chikungunya virus to mimic optimal human codon usage could, at least in part, be driven by competing selection to mimic mosquito codon usage. The authors didn't fully address my previous comment on this and, so far as I can see, there is also no mention of this issue in the manuscript. I feel that it is essential to at least mention this issue in the manuscript to give a proper background to understanding the possible reasons why CHIKV codon usage is suboptimal.

We apologize for this oversight and have provided an answer below the next comment.

Moreover, the authors didn't address the part of my previous comment where I asked "Can the authors calculate the CAI of CHIKV relative to the codon usage of the translationally activated host genes (i.e. is CHIKV CAI low because CHIKV is adapted to the stress-conditions codon optimality or is it low because it is adapted to mosquito cells, or a mixture?)." In the supplementary table showing codon usage in human, CHIKV and various other viruses, could the authors calculate a codon usage table for the human set of translationally activated genes, and then using this table as the baseline, calculate the CAI for all the rows in this table, and add this as another column. It will be important to see whether the CAI for CHIKV is

substantially increased when calculated relative to human translationally activated genes, and also how this compares with the CHIKV CAI relative to its mosquito host. Maybe it would also be useful to add a panel to Fig 4, comparing CHIKV vs Aedes?

We apologize for this oversight and have added a new paragraph to the results section containing the CAI of CHIKV RNA relative to the codon usage of translationally-activated human genes (CAI = 0.7288) and to the codon usage of *Aedes albopictus* overall (CAI = 0.7113), a known CHIKV vector. Moreover, we added a panel to Fig 4 comparing the codon usage of CHIKV and *Aedes albopictus*, and added the requested change in the supplementary table 5. These results show that CHIKV RNA does not appear to be more adapted to *A. albopictus* when compared to overall human (CAI = 0.7123), and that no particular enrichment for “rare” codons can be observed when comparing to all *A. albopictus* genes. However, one has to consider here that 1) this comparison does not take into account any gene expression changes induced by viral infection and 2) it is unclear whether the same tRNA modifications are required for efficient CHIKV translation in *A. albopictus*. Nonetheless, a recent study found that tRNA modification enzymes are expressed at higher levels in female mosquitoes of several species when compared to their male counterparts¹. Since only female mosquitoes are able to spread arboviruses via blood feeding (male mosquitoes do not blood feed), this finding opens an interesting avenue for further study of connections between tRNA modifications, viral RNA translation and arbovirus transmission. We have added these results to page 9.

MINOR COMMENTS:

- In the "Uncropped western blots" document, page 6, lower blot - I am curious as to why the marker lane (and some adjacent lanes) appear to disappear above 49 kDa? Could the greyscale be adjusted so the composition of this blot is more clear?

This WB was cut at 49 kDa to incubate the lower part with the rabbit Capsid antibody and the upper part first with the mouse tubulin and subsequently with the rabbit Nsp1 antibody. We used the following fluorescence-conjugated secondary antibodies: Rabbit emitting at 680nm and mouse emitting at 800nm. The western blot was detected using the Odyssey so both signals can be detected separately. In the previous version we showed only the 680nm “rabbit” channel, this is why the marker and tubulin could not be seen. We have exchanged the image so both channels are visible simultaneously and adjusted the grey scale.

- Are the data in Fig S3 averages over all libraries?

The averages were calculated using only samples that passed quality control as shown in Fig S26 C-D.

- In the Fig S3 caption, I found the meaning of "regions" unclear. The first sentence suggests the regions are 5'UTR, CDS and 3'UTR. However, this is not compatible with the later statement "... normalized by the total number of RPFs in the region", since in that case, in each normalized plot, the CDS mean value, 5'UTR mean value, and 3'UTR mean value should all be equal to each other. Instead, it looks like the "region" for

normalization is the full-length mRNA, so that full-length mRNA infected and full-length mRNA control have the same mean values.

We thank the reviewer for pointing out this issue. Indeed the normalization by total number of RPFs is performed on the entire transcript before creating a metagene with coordinates 0-100 for each individual region (5'UTR, CDS, 3'UTR). We have corrected this mistake in the caption of Fig S3.

- In Fig S3A and S3B, the x-axis scale is confusing and I did not see an explanation. Is it % of the way through the annotated 5'UTR, CDS or 3'UTR, respectively?

Yes, we indicated it now in the figure legend and also changed the labeling of the axis to make it clearer.

- In Fig S3, I assume it is P-site mapping, but this should be clarified in the caption.

Indeed Fig S3 displays P-site mapping profiles and we have clarified this point in the caption.

- In Fig S13-S16, the caption says P-site positions are being plotted. Please check the location of "0" and the red vertical line on each plot. It is a bit unusual that the main jump up in density from 5'UTR to CDS, and jump down in density from CDS to 3'UTR seems to occur a little offset from the red line (expect it to be one codon upstream at the stop codon since termination occurs with the stop codon in the A-site).

We thank the reviewer for pointing out this error. The peaks in these meta-profiles do not reflect the P-site positions themselves but the number of read extremities relative to the P-site (see Figure 1C in Lauria et al.²). We have corrected the y-axis labels and captions accordingly and included the reference to Lauria et al Figure 1C in the captions of FigS13-S16 (now FigS18-21).

- Lines 225-226: The authors say "Our study identifies an unanticipated virus-induced adaptation of the host translation machinery to the viral codon usage." I think it is wrong to say that the host has adapted to the viral codon usage. More likely that cell stress leads to reprogramming of codon optimality as a generic host response to deal with stress, and that viruses which cause cell stress have evolved their codon usage bias to reflect this endogenous reprogramming - i.e. the virus has adapted for optimal gene expression in the host infection environment.

We agree and have changed the text accordingly (page 13).

References:

1. Kelley, M. *et al.* Abundances of transfer RNA modifications and transcriptional levels of tRNA-modifying enzymes are sex-associated in mosquitoes. *Insect biochemistry and molecular biology* **143**, 103741 (2022).
2. Lauria, F. *et al.* riboWaltz: Optimization of ribosome P-site positioning in ribosome profiling data. *PLoS Comput Biol* **14**, e1006169 (2018).

Reviewer comments, third round –

Reviewer #3 (Remarks to the Author):

I thank the authors for carefully addressing my previous comments.

The use of lncRNAs (Supp Fig 4b) to test for contamination is nice - provided the transcript abundances (RNA-seq RPKMs) are sufficiently high for sufficiently many of the ER-associated lncRNA transcripts to get robust measurements.

The comparison of 3'UTR and CDS length distributions test (Supp Fig S5-S8) seems OK (so long as the read length distributions are being determined by the nuclease trimming and not being restricted by subsequent steps in the experimental protocol such as gel size selection).

Given that the codon adaptive index of CHIKV relative to human (CAI = 0.7123) and *Aedes albopictus* (CAI = 0.7113) were quite similar, and the CAI of CHIKV relative to the translationally upregulated human genes was not actually very much higher (CAI = 0.7288), I would suggest that the statement "CHIKV RNA codon usage mimics that of translationally-activated RNAs." may be a bit of an overstatement and that "Thus, the suboptimal coding usage of CHIKV RNA in human cells could be driven by a competing selection to adapt to the mosquito codon usage. However, this does not seem to be the case, as the CAI of CHIKV RNA relative to the codon usage of human (CAI = 0.7123) and *Aedes albopictus* (CAI = 0.7113), a known CHIKV vector, are quite similar when considering all annotated protein-coding genes." does not quite 'hit the nail on the head'. I'd suggest that global suboptimality of the CHIKV genome may be determined by alternating hosts, whereas preference for some specific codons may be determined by the stress-related change in tRNA modification status? But of course it is up to the authors how they interpret these data.

Minor comment:

In the inserted text "vertebrates and mosquitoes, two hosts separated by one million years in evolution", I think the correct time scale is ~one billion not one million years ;-)

REVIEWERS' COMMENTS

Reviewer #3 (Remarks to the Author):

I thank the authors for carefully addressing my previous comments.

The use of lncRNAs (Supp Fig 4b) to test for contamination is nice - provided the transcript abundances (RNA-seq RPKMs) are sufficiently high for sufficiently many of the ER-associated lncRNA transcripts to get robust measurements.

We appreciate the author's continued concerns and have thus compared the overall distributions of TPM expression values between lncRNAs and mRNAs (Fig. 1). To be able to compare the kallisto-generated TPM values between different samples we applied a between-sample normalization via sleuth. The comparison includes the 6015 lncRNAs we presented in Supplementary Figure S4b as well as the 12220 mRNAs that were analyzed in this study (after filtering out lowly expressed mRNAs).

As expected, lncRNAs follow the same expression pattern as their coding counterparts while being generally expressed at lower levels. Depending on the infected ER sample and which TPM threshold one uses to define well-expressed RNAs, between 700 and 1800 of the 6015 lncRNAs would be considered as sufficiently highly expressed, which should suffice to make robust observations.

Figure 1: Distribution of transcripts per kilobase million (TPM) values for long non-coding RNAs (lncRNAs) and mRNAs across all sequenced samples. Since different TPM thresholds have been commonly applied in literature to define “expressed genes”, TPM values were binned into ranges to facilitate the assessment of the corresponding percentage of RNAs falling into a specific bin (range).

The comparison of 3'UTR and CDS length distributions test (Supp Fig S5-S8) seems OK (so long as the read length distributions are being determined by the nuclease trimming and not being restricted by subsequent steps in the experimental protocol such as gel size selection).

We agree.

Given that the codon adaptive index of CHIKV relative to human (CAI = 0.7123) and *Aedes albopictus* (CAI = 0.7113) were quite similar, and the CAI of CHIKV relative to the translationally upregulated human genes was not actually very much higher (CAI = 0.7288), I would suggest that the statement "CHIKV RNA codon usage mimics that of translationally-activated RNAs." may be a **bit of an overstatement** and that "Thus, the suboptimal coding usage of CHIKV RNA in human cells could be driven by a competing selection to adapt to the mosquito codon usage. However, this does not seem to be the case, as the CAI of CHIKV RNA relative to the codon usage of human (CAI = 0.7123) and *Aedes albopictus* (CAI = 0.7113), a known CHIKV vector, are quite similar when considering all annotated protein-coding genes." does not quite 'hit the nail on the head'. I'd suggest that global suboptimality of the CHIKV genome may be determined by alternating hosts, whereas preference for some specific codons may be determined by the stress-related change in tRNA modification status? But of course it is up to the authors how they interpret these data.

We agree and modified the text accordingly (page 9).

Minor comment:

In the inserted text "vertebrates and mosquitoes, two hosts separated by one million years in evolution", I think the correct time scale is ~one billion not one million years ;-)

We corrected it.